# The asymmetric effects of climate risk on higher-moment connectedness among carbon, energy and metals markets

Yuqin Zhou[1], Shan Wu [2] ✉, Zhenhua Liu[3] & Lavinia Rognone [4,5]

Climate change affects price fluctuations in the carbon, energy and metals markets through physical and transition risks. Climate physical risk is mainly caused by extreme weather, natural disasters and other events caused by climate change, whereas climate transition risk mainly results from the gradual switchover to a low-carbon economy. Given that the connectedness between financial markets may be affected by various factors such as extreme events and economic transformation, understanding the different roles of climate physical risk and transition risk on the higher-moment connectedness across markets has important implications for investors to construct portfolios and regulators to establish regulation system. Here, using the GJRSK model, time-frequency connectedness framework and quantile-on-quantile method, we show asymmetric effects of climate risk on connectedness among carbon, energy and metals markets, with higher impacts of climate physical risk on upward risk spillovers, and greater effects of climate transition risk on the downside risk of kurtosis connectedness.

Risks associated with climate change typically encompass two types: physical risks and transition risks[1]. Climate-induced physical risks primarily arise from acute or chronic events such as increasing temperatures, rising sea levels, intensified storms, floods, and wildfires, leading to potential damages and losses[2]. On the other hand, transition risks result from the gradual shift towards a low-carbon economy, encompassing climate policies changes, consumer preferences shifts, and the emergence of competitive green technologies. Compared to traditional disaster risks, climate change risks usually exhibit a longer time scale, a broader geographic scope, more intricate scientific attribution, and higher levels of uncertainty[3]. These have drawn the attention of numerous scholars to investigate the influence of climate change on various financial markets, including the stock market[4,5], bond market[6,7], and currency market[8,9].

Previous research findings have demonstrated that climate change exerts a substantial influence on price and volatility, as well as supply and demand dynamics in carbon trading, energy, and metal markets (Supplementary Note 1). These effects occur through the channels of both physical risk and transition risk[10,11]. In terms of physical risk, climate change, especially at extreme high and low temperatures, can lead to more energy consumption and higher carbon emission prices[12,13]. Because clean energy power generation is highly dependent on weather factors[14], extreme climate change has also exposed clean energy companies to physical risks, seriously affecting carbon prices[15,16]. On the other hand, climate change may force governments to formulate relevant environmental policies such as carbon reduction, changing the supply of carbon emissions rights, and thus affecting carbon prices[17,18]. Driven by the "carbon neutrality" goal, countries have been forced to reduce traditional energy consumption, expand clean energy markets, and develop low-carbon basic products such as solar panels and power cells, increasing the demand for metal minerals[19,20]. Conversely, Carbon emissions trading is one of the mainstream ways to address climate change[21]. Energy consumption and the mining of metal minerals will

[1]School of Economics and Management, Chongqing Normal University, Chongqing, China. [2]School of Finance, Nanjing University of Finance and Economics, Nanjing, China. [3]School of Economics and Management, China University of Mining and Technology, Xuzhou, China. [4]University of Edinburgh Business School, The University of Edinburgh, Edinburgh EH8 9JS, UK. [5]Alliance Manchester Business School, The University of Manchester, Manchester M15 6PB, UK. ✉e-mail: wushaniam@outlook.com

increase the carbon dioxide emissions[22,23], increasing the risk of climate change[24].

Overall, previous studies have sought to provide evidence about the impact of climate risk on specific financial markets, with limited work exploring the connectedness between markets. Nevertheless, the carbon, energy, and metal markets are closely linked[21,25]. Higher energy and metal prices will lower the demand for carbon dioxide emissions, leading to lower carbon prices[26,27]. Rising carbon prices will partly encourage companies to reduce energy consumption[28] and the proportion of metal mining such as raw ore[29], which affects the profitability of related companies such as metal energy[30]. Hence, climate change risk will not only have an impact on the carbon, energy and metal markets, respectively, but it is also expected to have an important effect on the connectivity of the whole system formed by different markets. For instance, Ding et al.[31] show an evident causal relationship between climate change and the spillover effects between carbon and energy markets. However, they mainly measure climate change from the perspective of investors' attention, and do not deeply take into account the heterogeneous effects of different risks of climate change on the correlation between markets.

Meanwhile, the objectives and preferences of market participants differ in the short and long term, leading to variations in spillover effects among carbon, energy, and metal markets at different frequencies[23]. The price behaviors of these markets also exhibit various frequencies, ranging from long-term trends to short-term fluctuations. The cap-and-trade principle of the European Union Emissions Trading System affects the short- and long-term price of carbon market trading demand, causing differences in frequency spillover effects[31,32]. Furthermore, energy and metal commodity markets have sticky prices in the short term and exhibit complete flexibility in the long term, resulting in different frequency connections with other markets[33]. In addition, the existing literature primarily focuses on the spillover effects in terms of return and volatility (Supplementary Note 2), while the literature on the impact of climate risk on higher-order moment risk remains limited. However, higher moment risks on asset pricing, volatility modeling and risk hedge and portfolio optimization are important. Many studies suggest that the interrelationship between different financial assets is reflected in the second moment and higher moments[34,35].

In this work, we employ the GJRSK model proposed by Nakagawa and Uchiyama[36], the spillover method introduced by Baruník and Krehlík[37], and the quantile-on-quantile method to investigate the influence of climate risk on high-order time-frequency spillover effects. First, we explore the risk spillover effects in the carbon-energy-metals nexus from the perspective of higher-order moment risk, adding to the previous studies, which mainly investigate the spillover effects of returns and volatility. Our results provide evidence that the risk spillover effects at the higher-order moment level should be considered, while the spillover effects of skewness or kurtosis risk among carbon, energy, and metal markets are lower compared to volatility risk spillover. Second, the high-order moment risk spillovers are decomposed in different frequency domains, and combined with multidimensional network analysis, the high-order moment risk transition path in the carbon-energy-metals nexus and its heterogeneity in the short-term and long-term frequency domains are characterized. We find that risk spillovers show strong heterogeneity at different frequencies and higher-order moment levels, with the long-term volatility spillovers accounting for the largest proportion of cross-market volatility spillovers, while high-order moment risk spillovers mainly occur in the short term. Finally, this study examines the impact of climate risk on the risk spillover effects in the carbon-energy-metals nexus via measures of physical and transition climate risk proxies obtained from textual analysis. The empirical findings demonstrate that both physical and transition risks positively affect short-term kurtosis spillovers, but have a negative impact on long-term volatility

and skewness spillover effects. Our findings will provide valuable reference evidence for investors and policymakers in devising risk management strategies and making informed investment decisions.

## Results and discussion

To investigate the time-frequency high-order risk spillover effects among carbon, energy, and metal markets, we employ the spillover method introduced by Baruník and Křehlík[37] along with the GJRSK model. By combining network analysis, we can unveil the network structure and transmission pathways of time-frequency high-order risk spillover in the carbon-energy-metal nexus. This approach aids in determining the relative significance of each market within the system. Moreover, we employ the quantile-on-quantile method to examine the influence of climate risk on the overall spillover effects within the carbon-energy-metal markets.

### Static spillover effects

We employ the GJRSK model to estimate the conditional variance, skewness, and kurtosis of each market. The parameter estimation results of the GJRSK model (Supplementary Table 2) show that the leverage coefficient in the equations for conditional variance, skewness and kurtosis is observably non-zero, indicating a striking leverage effect on the conditional volatility, skewness and kurtosis in the carbon, energy and metal future markets, which illustrates that the GJRSK model is better suited to measuring higher order moments risk of carbon, energy, and metal markets than the GARCHSK model. The $\beta_3$ in each market is evidently greater than 0, indicating that negative impacts in the carbon, energy and metal markets will lead to greater fluctuations or more extreme events than the same degree of positive impacts. This may be because bad news in financial markets often have a more negative impact on asset prices than the positive impact of good news. The kurtosis leverage effect ($\delta_3$) is positive, indicating that the negative return with a large absolute value often appears, which will cause the probability of the extreme value of the return to appear greater than the probability of the extreme value under the normal distribution, thus increasing the kurtosis of the return distribution. In contrast, the skewed leverage effect ($\gamma_3$) of most markets is evidently positive, indicating that the frequency and magnitude of negative returns in these markets are often greater than positive returns, except for oil, gold, and zinc markets.

Supplementary Fig. 1 displays the time-varying conditional variance, skewness, and kurtosis of each market. It provides insights into how these measures fluctuate over time and highlights the dynamic nature of market conditions. The variations in conditional volatility, skewness, and kurtosis observed across different markets highlight the distinctive information processing characteristics to each market category. Dynamic volatility, skewness and kurtosis can stem from actual facts. For example, the outbreak of COVID-19 in January 2020[23] led to sharp fluctuations of the price of Oil, Gold, Silver and other markets, or, as a consequence of the escalated conflict between Russia and Ukraine in February 2022[38], the supply of energy and non-ferrous metals market was affected, resulting in relatively large price fluctuations of Gas, Coal, Nickel and Zinc markets.

To enhance our understanding of time-frequency spillovers in the carbon-energy-metals nexus, we apply the spillover index approach developed by Diebold and Yilmaz[39,40] (DY model) and the frequency connectedness method proposed by Barunik and Krehlik[37] (BK model). These models enable us to effectively measure the connectedness within and across different frequencies of these markets. In this study, the lag order of the VAR model has been determined to be 2 based on the AIC criterion. Additionally, a fixed forecast horizon of 100 days has been utilized for the analysis. In the frequency domain framework, two distinct timescales are taken into consideration: the high frequency, ranging from 1 to 22 days, and the low frequency, encompassing durations greater than 22 days.

**Table 1 | Static variance spillovers using the DY model (%)**

|  | EUA | Oil | Gas | Coal | Gold | Silver | Copper | Aluminum | Zinc | Nickel | Tin | Lead | FROM |
|---|---|---|---|---|---|---|---|---|---|---|---|---|---|
| EUA | 78.25 | 6.04 | 1.35 | 3.38 | 0.43 | 2.12 | 1.57 | 0.76 | 0.38 | 1.10 | 3.36 | 1.26 | 1.81 |
| Oil | 6.16 | 69.54 | 0.45 | 1.20 | 4.09 | 7.59 | 5.48 | 0.60 | 0.75 | 0.26 | 2.91 | 0.97 | 2.54 |
| Gas | 7.39 | 0.65 | 56.36 | 14.49 | 0.74 | 0.37 | 2.77 | 0.78 | 3.60 | 2.10 | 7.93 | 2.83 | 3.64 |
| Coal | 9.62 | 0.94 | 19.44 | 51.68 | 1.85 | 0.65 | 5.95 | 0.06 | 2.33 | 0.29 | 5.07 | 2.12 | 4.03 |
| Gold | 5.10 | 26.82 | 0.71 | 0.96 | 25.21 | 36.04 | 2.18 | 0.35 | 0.60 | 0.50 | 0.82 | 0.71 | 6.23 |
| Silver | 0.52 | 7.71 | 0.49 | 0.30 | 14.35 | 68.56 | 1.95 | 0.57 | 2.02 | 0.44 | 0.74 | 2.35 | 2.62 |
| Copper | 4.41 | 18.55 | 2.11 | 0.43 | 0.48 | 16.00 | 38.77 | 1.91 | 5.76 | 1.49 | 8.53 | 1.55 | 5.10 |
| Aluminum | 4.79 | 0.54 | 5.91 | 12.72 | 1.88 | 0.51 | 5.29 | 60.23 | 3.49 | 0.84 | 2.71 | 1.09 | 3.31 |
| Zinc | 5.08 | 0.33 | 13.53 | 14.67 | 1.06 | 0.22 | 12.60 | 3.13 | 33.76 | 0.92 | 2.97 | 11.73 | 5.52 |
| Nickel | 4.92 | 0.35 | 12.29 | 19.42 | 0.70 | 0.51 | 2.37 | 0.86 | 3.59 | 46.18 | 7.51 | 1.30 | 4.49 |
| Tin | 4.68 | 2.06 | 13.23 | 7.59 | 1.58 | 1.00 | 3.90 | 7.81 | 9.44 | 0.51 | 41.53 | 6.66 | 4.87 |
| Lead | 2.20 | 0.41 | 2.04 | 2.58 | 0.86 | 2.20 | 1.93 | 2.05 | 15.59 | 1.06 | 7.54 | 61.55 | 3.20 |
| TO | 4.57 | 5.37 | 5.96 | 6.48 | 2.33 | 5.60 | 3.83 | 1.57 | 3.96 | 0.79 | 4.17 | 2.72 | TCI = 47.37 |

The table presents the static spillover connectedness based on the DY method. We provide the total spillover index (denoted by the term "TCI"), the directional spillover received (denoted by "FROM"), and transmitted (denoted by "TO") by each market. The *jk* th value is the directional connectedness from *k* to *j*.

The outcomes of volatility spillovers among carbon, energy, and metals markets are presented in Table 1, utilizing the DY model. Table 1 shows that the overall volatility connectedness index for the system is 47.37%. Coal emerges as the dominant transmitter of volatility spillovers, contributing 6.48% to other markets, while the gold market takes the forefront as the largest receiver, absorbing 6.23% from other markets. The carbon market had the highest spillover to the Coal market (9.62%) which had the largest spillover to the Nickel market (19.42%), and the EUA accepted the largest spillover from the Oil market which accepted the biggest spillover from the Silver market, reflecting the connectivity among the carbon, energy and metals markets. However, it is worth noting that carbon markets exhibit a relatively low level of receptiveness, which is consistent with the findings of Jiang and Chen[23] and Qi et al.[41], suggesting that carbon market volatility has relatively little impact on the dynamics of the carbon-energy-metal market system. The results of the BK model's volatility spillover analysis (as shown in Table 2) indicate that the total volatility spillover indices at high-frequency and low-frequency bands are 6.03% and 41.34%, respectively. Clearly, the magnitude of volatility spillovers in the short run, characterized by high frequency, is comparatively lower in comparison to the long run, characterized by low frequency. In the short run, the Zinc market stands out as the principal transmitter of volatility, while the Tin market assumes the role of the primary receiver. In the long run, the Coal market emerges as the primary transmitter of volatility, while the Gold market demonstrates the highest susceptibility as a receiver of volatility. At any frequency, the volatility of carbon price is more affected by the price fluctuations of the Oil market, while the high-carbon emission industries should also focus on the Copper and Tin markets in the short and long run. The spillover of EUA leads to fluctuations of the Oil and Coal markets in the short and long run, respectively. Overall, the carbon markets are more correlated to energy markets. The Supplementary Table 3 provides the results of skewness spillovers obtained from both the DY and BK models, encompassing twelve different markets. The total skewness connectedness index for this system is observed to be 12.08%, suggesting a relatively weak interdependence in terms of skewness among these markets. And each market is predominantly influenced by its own shocks rather than the skewness of other markets. Among the considered markets, Tin emerges as the primary recipient of skewness spillover, while Aluminum stands out as the main transmitter of skewness spillover. The Lead and Copper markets exhibit the highest skewness spillovers among all market pairs, indicating a strong transmission of skewness information between these two markets. The results of the BK model reveal that the total skewness spillover index

amounts to 10.89% in the high-frequency band and 1.19% in the low-frequency band. This indicates that skewness spillovers have a more significant impact in the higher frequency range as opposed to the lower frequency range, indicating that the skewness spillovers among these markets is primarily driven by short-term effects. The results of kurtosis connectedness derived from both the DY and BK models are presented in Supplementary Table 4. The overall kurtosis spillover index for the entire system amounts to 20.01%, with 17.12% and 2.89% observed at high-frequency and low-frequency bands, respectively. The results indicate that, similar to the findings for skewness spillovers, the level of kurtosis spillovers is higher at high frequencies compared to low frequencies. The Aluminum market takes on the role of the primary transmitter of kurtosis, while the Zinc market serves as the major receiver of kurtosis. In the long run, Silver emerges as the primary transmitter of kurtosis, while Gold assumes the position of being the most significant recipient of kurtosis.

In summary, the total spillover indices for volatility, skewness, and kurtosis in the time domain are recorded as 47.37%, 12.08%, and 20.01%, respectively. These results demonstrate an inconsistency with the findings reported by Yang et al.[35], who detect a weakening of system connectedness as the order of moments increased. However, we find that the spillover of kurtosis is not necessarily lower than the spillover of skewness. The findings indicate that volatility spillover effects exhibit the highest intensity. This is consistent with the assertion made by Bouri et al.[11], who argue that volatility spillovers tend to be more influential in terms of the strength of linkages among markets.

Moreover, the spillovers across carbon, energy and metals markets have a clear difference at different moments. For example, the Silver and Oil markets have a relatively high volatility connectivity to the gold market, while the skewness connectivity between the Lead and Copper markets and the kurtosis connectivity between the Aluminum and Zinc markets contributed more to the total skewness and kurtosis spillover, respectively. The above results demonstrate the importance of considering the contagion effects between markets from higher-order moments. From a frequency domain perspective, long-term volatility connectedness contributes more to the overall volatility spillover, which contradicts findings in several studies[23,35]. However, for skewness and kurtosis, high-frequency band connectedness plays a dominant role in their respective total spillovers. This suggests that skewness and kurtosis information transmits rapidly in the nexus among carbon, energy, and metal markets. This is due to the fact that skewness and kurtosis spillover effects are predominantly driven by market participants with shorter investment durations and

**Table 2 | Static variance spillovers using the BK model (%)**

| | EUA | Oil | Gas | Coal | Gold | Silver | Copper | Aluminum | Zinc | Nickel | Tin | Lead | FROM |
|---|---|---|---|---|---|---|---|---|---|---|---|---|---|
| Panel A: Frequency 1 (High frequency): 1day to 22 days | | | | | | | | | | | | | |
| EUA | 34.68 | 0.95 | 0.29 | 0.75 | 0.32 | 0.49 | 0.80 | 0.27 | 0.03 | 0.19 | 0.31 | 0.60 | 0.42 |
| Oil | 0.75 | 15.47 | 0.02 | 0.02 | 0.12 | 0.06 | 2.21 | 0.30 | 0.29 | 0.03 | 0.72 | 0.49 | 0.42 |
| Gas | 0.43 | 0.03 | 20.05 | 0.72 | 0.05 | 0.02 | 0.07 | 0.51 | 0.91 | 0.33 | 2.30 | 0.66 | 0.50 |
| Coal | 0.11 | 0.01 | 0.48 | 3.50 | 0.02 | 0.02 | 0.14 | 0.04 | 0.01 | 0.03 | 0.03 | 0.02 | 0.08 |
| Gold | 0.07 | 0.32 | 0.00 | 0.01 | 1.87 | 0.29 | 0.12 | 0.05 | 0.06 | 0.07 | 0.04 | 0.05 | 0.09 |
| Silver | 0.08 | 0.20 | 0.03 | 0.01 | 2.39 | 8.41 | 0.19 | 0.03 | 0.10 | 0.08 | 0.04 | 0.05 | 0.27 |
| Copper | 0.07 | 0.14 | 0.02 | 0.03 | 0.02 | 0.06 | 1.79 | 0.13 | 0.25 | 0.17 | 0.22 | 0.23 | 0.11 |
| Aluminum | 0.35 | 0.10 | 0.09 | 0.84 | 0.34 | 0.07 | 1.01 | 41.53 | 2.36 | 0.35 | 1.36 | 0.56 | 0.62 |
| Zinc | 0.04 | 0.00 | 0.03 | 0.41 | 0.16 | 0.04 | 0.01 | 0.33 | 1.99 | 0.12 | 0.78 | 0.69 | 0.22 |
| Nickel | 0.16 | 0.02 | 0.48 | 0.20 | 0.10 | 0.02 | 0.32 | 0.38 | 0.76 | 20.74 | 1.37 | 0.19 | 0.33 |
| Tin | 0.44 | 0.12 | 1.28 | 0.54 | 0.83 | 0.34 | 0.59 | 4.65 | 5.11 | 0.26 | 20.33 | 3.60 | 1.48 |
| Lead | 0.53 | 0.05 | 0.27 | 0.27 | 0.37 | 0.99 | 1.13 | 1.42 | 8.11 | 0.71 | 4.15 | 33.58 | 1.50 |
| TO_ABS | 0.25 | 0.16 | 0.25 | 0.32 | 0.39 | 0.20 | 0.55 | 0.68 | 1.50 | 0.20 | 0.94 | 0.60 | TCI = 6.03 |
| Panel B: Frequency 2 (Low frequency): 22 days to infinity | | | | | | | | | | | | | |
| EUA | 43.57 | 5.09 | 1.06 | 2.63 | 0.11 | 1.63 | 0.77 | 0.48 | 0.36 | 0.91 | 3.05 | 0.66 | 1.40 |
| Oil | 5.40 | 54.08 | 0.44 | 1.18 | 3.97 | 7.53 | 3.27 | 0.30 | 0.46 | 0.23 | 2.18 | 0.47 | 2.12 |
| Gas | 6.96 | 0.62 | 36.31 | 13.76 | 0.69 | 0.35 | 2.69 | 0.27 | 2.69 | 1.77 | 5.63 | 2.17 | 3.13 |
| Coal | 9.51 | 0.94 | 18.97 | 48.18 | 1.82 | 0.63 | 5.81 | 0.03 | 2.32 | 0.25 | 5.03 | 2.11 | 3.95 |
| Gold | 5.03 | 26.51 | 0.71 | 0.95 | 23.34 | 35.75 | 2.06 | 0.30 | 0.54 | 0.43 | 0.78 | 0.67 | 6.14 |
| Silver | 0.45 | 7.51 | 0.45 | 0.29 | 11.96 | 60.14 | 1.77 | 0.54 | 1.92 | 0.36 | 0.70 | 2.30 | 2.35 |
| Copper | 4.34 | 18.41 | 2.09 | 0.41 | 0.46 | 15.94 | 36.99 | 1.79 | 5.51 | 1.32 | 8.31 | 1.32 | 4.99 |
| Aluminum | 4.44 | 0.44 | 5.82 | 11.88 | 1.55 | 0.44 | 4.28 | 18.70 | 1.13 | 0.49 | 1.35 | 0.53 | 2.70 |
| Zinc | 5.05 | 0.33 | 13.50 | 14.26 | 0.90 | 0.18 | 12.58 | 2.80 | 31.78 | 0.80 | 2.19 | 11.04 | 5.30 |
| Nickel | 4.76 | 0.33 | 11.81 | 19.22 | 0.60 | 0.49 | 2.05 | 0.48 | 2.83 | 25.43 | 6.14 | 1.11 | 4.15 |
| Tin | 4.24 | 1.94 | 11.94 | 7.05 | 0.75 | 0.66 | 3.31 | 3.17 | 4.33 | 0.25 | 21.20 | 3.06 | 3.39 |
| Lead | 1.68 | 0.36 | 1.77 | 2.31 | 0.49 | 1.21 | 0.80 | 0.63 | 7.48 | 0.35 | 3.39 | 27.97 | 1.70 |
| TO_ABS | 4.32 | 5.21 | 5.71 | 6.16 | 1.94 | 5.40 | 3.28 | 0.90 | 2.46 | 0.60 | 3.23 | 2.12 | TCI = 41.34 |

The table presents the static spillover connectedness based on the BK model. We provide the total spillover index (denoted by the term "TCI"), the directional spillover received (denoted by "FROM"), and transmitted (denoted by "TO_ABS") by each market. The $jk$ th value is the directional connectedness from $k$ to $j$.

trading strategies. These findings contrast with the conclusions drawn by Bouri et al.[42], whose research indicated that spillover levels in all implied moments were notably higher at lower frequencies. This discrepancy may stem from the distinct characteristics of conditional and implied higher-order moments, where the latter reflects market participants' expectations regarding future market behavior[43]. The metal market exhibits significant transmitters and receivers across multiple dimensions, highlighting its crucial role within carbon-energy-metal systems. This finding aligns with the conclusions drawn by Jiang and Chen[23]. This phenomenon could be attributed to the close relationship between metal production, high carbon emissions, and energy consumption. Metals are essential raw materials in clean energy technologies, which are rapidly being developed to reduce carbon emissions[44]. Therefore, metal prices are crucial to the price of energy and carbon trading.

**Rolling-windows analysis**

One significant drawback of the static spillover index lies in its underlying assumption that the relationships of volatility, skewness, and kurtosis between carbon, energy, and metals markets remain constant over time[45]. Some economic and financial events might have taken place during the sample period and have influenced the dependence across these markets. Figure 1 displays the dynamics of the total volatility, skewness, and kurtosis spillover indices, which were derived using a rolling window of 200 days. This implies that we have computed these indexes for 1452 different time periods, spanning from April 25, 2016 to June 30, 2022.

A careful examination of Fig. 1 reveals that the total volatility, skewness, and kurtosis spillover effects exhibit obvious variations over time. The overall volatility, skewness, and kurtosis spillover index experienced a spike after significant events such as the trade disputes between China and the United States in 2008, the outbreak of the COVID-19 epidemic in 2020, and the conflict between Russia and Ukraine in 2022. The amplification of spillovers across carbon, energy, and metals markets during extreme event shocks provides evidence of financial risk contagion. This finding aligns with the conclusions drawn from prior studies[25]. Skewness and kurtosis spillovers exhibit higher sensitivity to market information compared to volatility. These results are in line with existing literature[21,27]. Moreover, the observed volatility spillover effect in the high-frequency band is always lower compared to the low-frequency band. This finding is inconsistent with the results presented by Jiang and Chen[23], who provide evidence that the primary driver contributing to overall spillover in carbon, energy, and metals markets is predominantly the short-term spillover effects. This could be due to varying spillover effects at different frequencies on return and volatility. While the higher skewness and kurtosis spillover effects observed at higher frequency bands suggest that short-term shocks play a significant role in driving the total skewness and kurtosis spillover effects across carbon, energy, and metals markets. It is worth noting that many spillover peaks appear in the long and short term. This implies that risk events not only induce short-term market volatility but also have lasting impacts, persisting for more than a month and leading to significant spillover effects in both the short and long term. Compared to the skewness and kurtosis, the total volatility

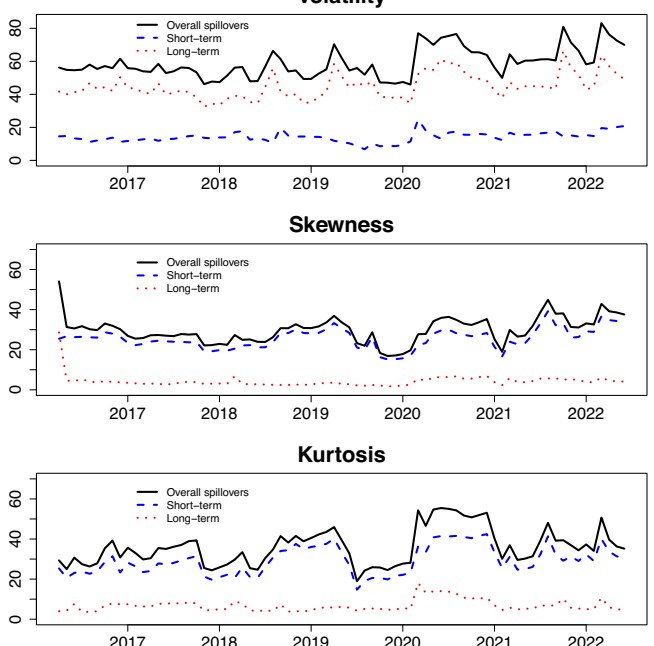

**Fig. 1 | Dynamic overall spillovers of carbon, energy and metals markets.** In order to avoid extreme values from masking other period trends, we drew the plots with the monthly mean of dynamic total connectedness. The top panels "Overall spillovers" is the dynamic total spillover index of the DY model. "Short-term" and "Long-term" is the dynamic frequency connectedness on the band:3.14 to 0.14 and 0.14 to 0 of BK model, respectively.

spillover effects have many obvious peaks, indicating that the dynamic connectivity of volatility is more volatile.

To gain deeper insights into the dynamic nature pertaining to spillover effects within the carbon, energy, and metal markets, our study focuses on examining the net directional spillover observed in each respective market. This analysis allows us to understand the flow and intensity of spillovers between these markets over time. Figure 2 showcases the dynamic estimation of net volatility spillover indices. These indices represent the discrepancy between the volatility transmitted from a particular market to the overall system and the volatility received by that particular market from the system. In summary, a positive net spillover index indicates the transmission of shocks from the market to other markets, whereas a negative net spillover index suggests the reception of shocks from other markets. It is evident that the net volatility spillovers demonstrate significant variability over time, with the EUA and Oil markets consistently acting as contributors of shocks throughout most of the sample period. However, the net volatility spillovers of most metal markets burst in either a negative or a positive direction. Due to the outbreak of the COVID-19 epidemic, Oil prices have fluctuated sharply, with net volatility spillover effects reaching 20% in March 2020. Then, the Zinc market experienced a net volatility spillover effect over 8% in October 2021. This surge was primarily triggered by Nyrstar's decision to reduce zinc production, which subsequently set off a series of events leading to a remarkable increase in industrial metal prices.

Supplementary Fig. 2 shows the dynamic net skewness time-frequency spillover effects. With the exception of extreme spillovers, the range of net skewness spillover effects is comparably narrower when compared to net volatility spillover. As can be seen from Supplementary Fig. 2, EUA, Aluminum, and Lead play a role as transmitters of skewness information, while Oil, Silver, and Zinc are more inclined to receive such information. During the observed period, the Aluminum market predominantly exhibits positive net skewness spillover,

while the Silver market primarily shows negative net skewness spillover. These findings indicate that the Aluminum market plays a dominant role as a transmitter of net skewness spillovers within the carbon-energy-metal system. Supplementary Fig. 3 provides a visual representation of how net kurtosis spillovers vary over time and across different frequency bands. The Gas market largely serves as a transmitter of net kurtosis, while the Oil market is primarily a receiver of net kurtosis. In particular, during the China-US trade tensions in 2018, there were observable positive spillover effects of Gas on other markets. In 2016, the net kurtosis spillover effects of the carbon market surpassed 5% due to the increased economic uncertainty resulting from the UK's exit from the EU. In summary, the findings of the net spillover effects demonstrate a clear manifestation of higher-order moment risk spillovers across carbon, energy, and metals markets.

We contrast the net spillover effects before and after the COVID-19 outbreak on 1 January 2020[23]. Figure 3 reports the average net total directional connectedness across carbon, energy, and metal markets in the full-sample period and two phases. The COVID-19 epidemic period witnessed a significant increase in absolute net directional connectedness across most markets, particularly in terms of net skewness and kurtosis spillovers. This surge in connectedness indicates a heightened propagation of risk and uncertainty during the pandemic compared to the pre-pandemic period. The influence of COVID-19 on risk spillover is widespread across various domains. This may be because the negative sentiment caused by the COVID-19 epidemic may trigger pessimistic expectations among investors in the market and amplify the spillover effects between markets through asset adjustments. Additionally, we can observe how the roles of each market change across different periods. Volatility in the Gold and Copper markets was mainly net spillovers before the outbreak of COVID-19, but converted to net recipients during the COVID-19 epidemic. However, the results from the COVID-19 outbreak period indicate that Gold and Copper markets exhibit positive short-term net volatility connectedness, aligning with the findings reported by Jiang and Chen[23].

## Network analysis of time-frequency spillover

To examine the network of net-pairwise directional connectedness among the analyzed markets, we construct a connectivity network of volatility, skewness, and kurtosis over both the time and frequency domains. To mitigate the influence of external extreme events, such as the Russia-Ukraine conflict, on the spillover network, we utilize the average of dynamic net-pairwise directional connectedness observed across the whole sample period to construct and visualize the network, as depicted in Fig. 4. Figure 4 demonstrates that the network structure of spillover effects in the carbon-energy-metals nexus exhibits distinct features across various dimensions. The location of the nodes is decided by the algorithm of Fruchterman and Reingold[46], which is one of the most used layout algorithms[47] and applies an iterative process to select the location of the nodes to minimize the energy of the system. Then, the nodes sharing more connections are located closer to each other.

Upon closer examination of Fig. 4, we observe that Gas, Aluminum, Silver, Coal, and Copper are the primary volatility transmitters, while EUA, Oil, Lead, Gold, Tin, and Zinc are the major volatility recipients, as indicated by the results of the DY model. According to the BK results, Gas exerts significant influence in the volatility connectedness, acting as a spillover transmitter of the other 10 markets. On the other hand, the Coal market tends to receive volatility from the other markets in the short run. In the longer run, Silver shows stronger volatility spillover effects. The key net transmitters in both the time-domain and high-frequency skewness connectedness networks are Silver, Gold, Oil, Copper, and Aluminum, whereas Lead, Nickel, and Tin act as net receivers. In a low-frequency skewness connectedness network, Gold undergoes a role transformation and becomes a receiver, indicating

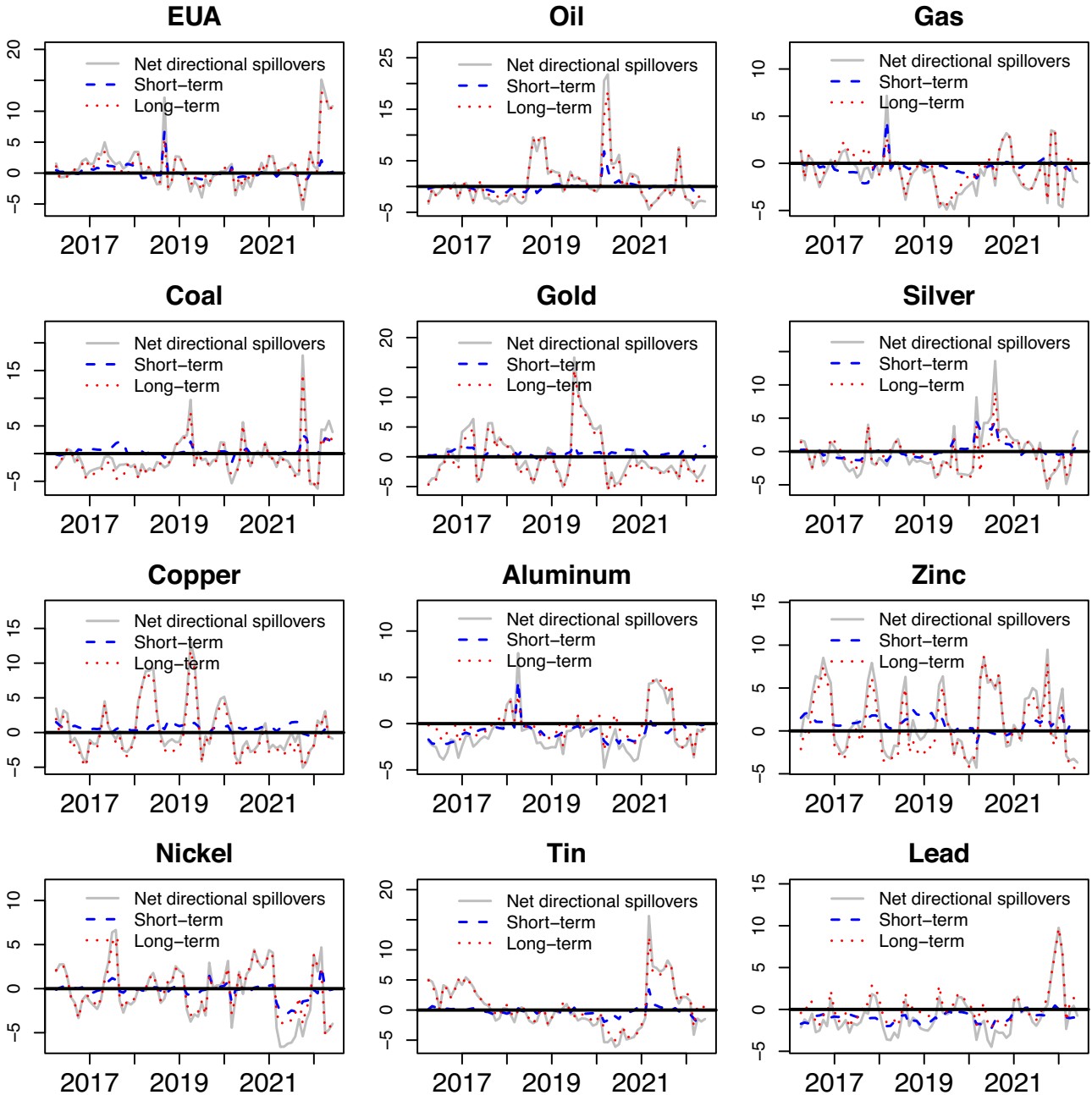

**Fig. 2 | Dynamic net volatility directional spillovers.** The plots are drawn by the monthly mean of dynamic net directional spillovers. The black horizontal line represents y = 0. "Net directional spillovers" is the aggregate net volatility directional spillovers of the DY model. "Short-term" and "Long-term" is the net volatility directional spillover in the short and long-term horizons of BK model, respectively. The top panels represent each market.

that the transmission of skewness shocks by gold primarily occurs in the short term. In the kurtosis spillover network, Oil, Copper, and Aluminum serve as net transmitters to other markets, while Lead, Nickel, and Zinc are net receivers. Over an extended period, Gas demonstrates more pronounced spillover effects in terms of kurtosis. These findings suggest that the energy and metal markets play a leading role in carbon-energy-metal systems. However, the roles of various markets in different higher-order moment risk spillovers are inconsistent. For instance, the Oil market serves as a transmitter of volatility risk spillover and also exhibits a susceptibility to skewness and kurtosis risks.

Supplementary Figs. 4 and 5 provide visual representations of the net pairwise directional spillover during both the pre and during COVID-19 periods. The findings indicate that the carbon market is

vulnerable to shocks in other markets, demonstrating higher vulnerability both in the short and long run. The magnitude of spillover effects between metal markets and energy markets is comparatively higher, indicating that these markets exert a more substantial influence and play a dominant role in the overall dynamics. Importantly, it should be emphasized that the spillover effect between the carbon market and the energy and metal markets has experienced an intensification amidst the COVID-19 pandemic period, demonstrating the heightened significance of the carbon market within carbon-energy-metal systems.

To enhance the representation of the relative significance of each market within the network, we introduce additional measures such as closeness centrality, betweenness centrality, and Pagerank centrality[48]. These measurements provide useful insights into the

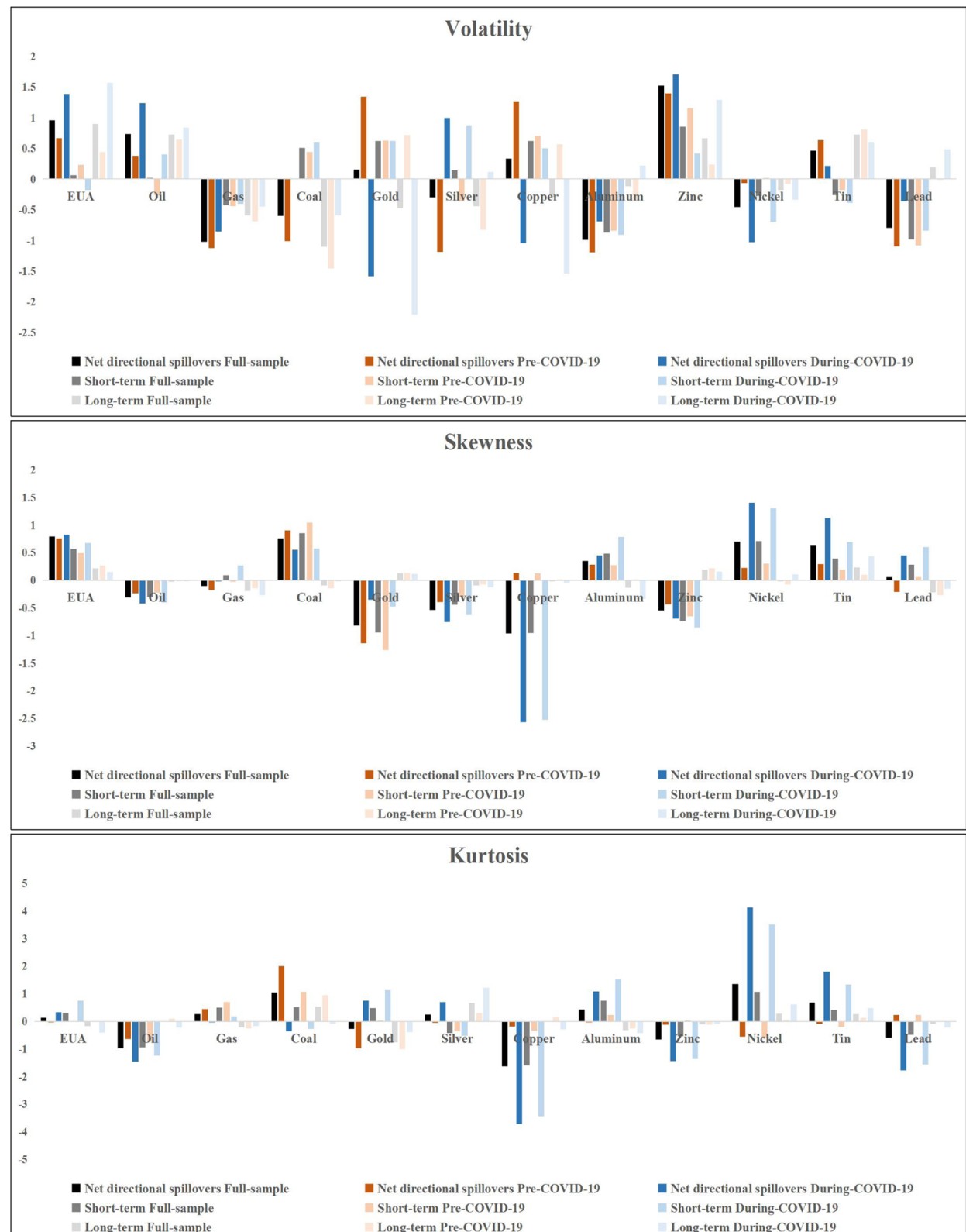

**Fig. 3 | Net total directional connectedness.** This figure shows the net total directional connectedness, representing the mean of each period's dynamic net total directional connectedness. The top panels represent net total directional connectedness of different order moments.

importance and impact of individual markets within the overall network structure. Closeness centrality measures the average distance or proximity between a given node and all other nodes in a network. The closeness centrality of node $j$ is computed using formula (1), where $C_{ij} \equiv \min_{C}\{C \in [1, N-1] : (j \xrightarrow{C} i) = 1\}$ is the length of the

shortest path from $j$ to $i$. According to Barthelemy[49], the betweenness centrality ($BC(v)$) of node $v$ can be defined as Eq. (2), where $R_1$ and $R_2$ represent two regions, $\sigma_{st}$ represents the total number of shortest paths from a specific node $s$ to another specific node $t$, and $\sigma_{st}(v)$ refers to the number of shortest path from $s$ to $t$ traverse via a

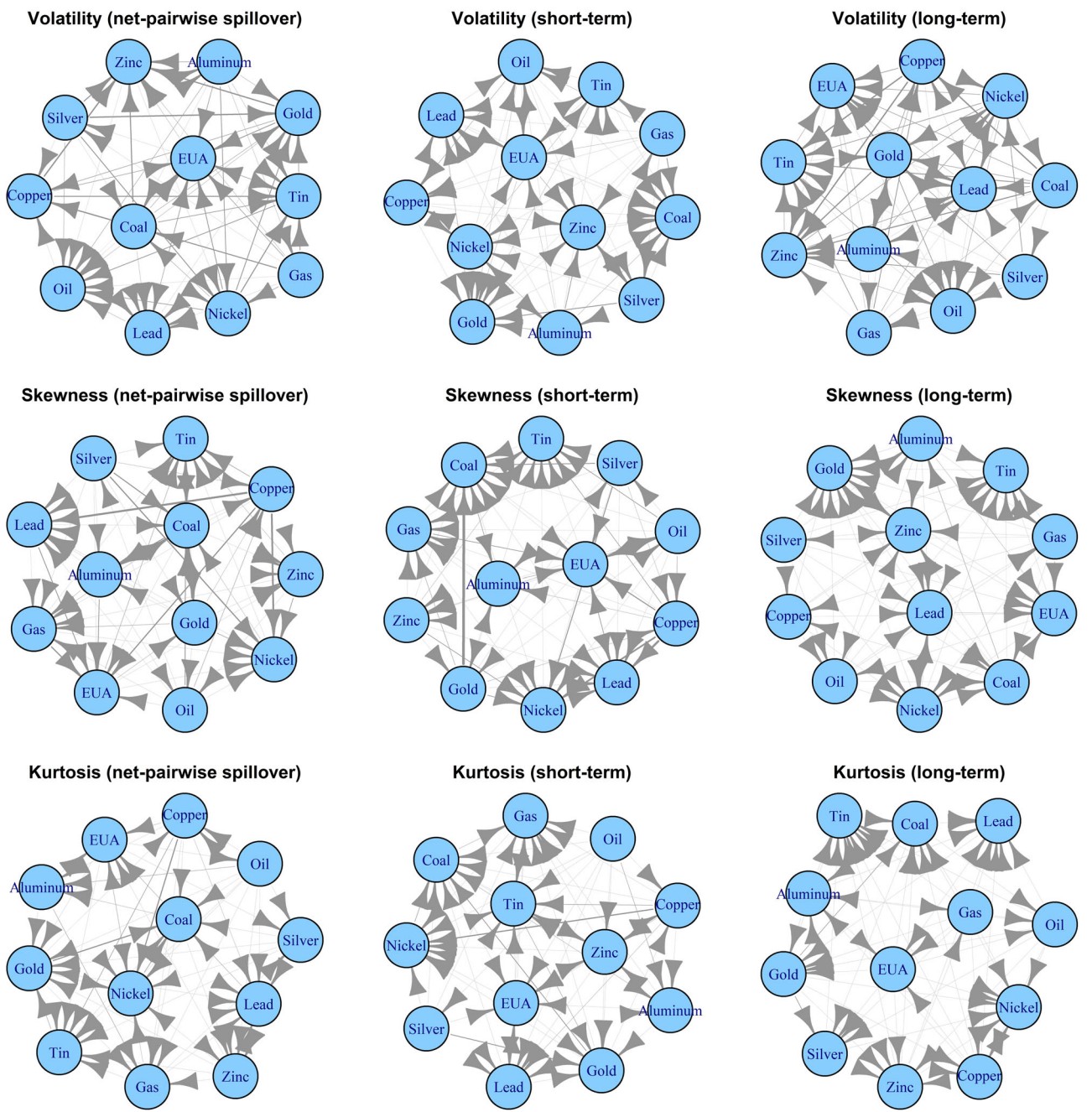

**Fig. 4 | Net-pairwise directional connectedness in the full-sample period.** This figure shows the 66 pairs of carbon, energy and metals markets. The nodes represent each market, and the thickness of the edge shows the degree of the net-pairwise directional connectedness. The arrows going from markets $i$ to $j$ represent net spillovers, that is, the contribution of market $i$ to market $j$ is greater than that of market $j$ to market $i$. "Net-pairwise spillovers" is the aggregate net-pairwise directional spillovers of the DY model. The "short-term" and "long-term" is the net-pairwise directional spillover in the short and long-term horizons of BK model, respectively. The top panels represent the volatility, skewness and kurtosis network in different frequencies, such as "Volatility (short-term)" represents the network drawn by the net-pairwise directional volatility connectedness in the short-term horizons of BK model in the full-sample period.

specific node $v$. If a member of a network lies on multiple shortest paths connecting other members, it can be considered the core member with higher betweenness centrality.

$$C_{js} = \frac{1}{N-1} \sum_{i \neq j} C_{ji} \left( j \xrightarrow{C} i \right) \qquad (1)$$

$$BC(v) = 2 \sum_{\substack{s \in R_1, t \in R_2 \\ s \neq v \neq t}} \frac{\sigma_{st}(v)}{\sigma_{st}} \qquad (2)$$

Eigenvector centrality measures the significance of a node within a network by assigning relative scores based on its connectivity with other nodes. The higher the degree of connection a node has with other nodes in the network, the more important it is considered. The eigenvector centrality of $j$ is the sum of eigenvector centrality of all nodes that are infected by $j$: $V_j = \sum_{i=1}^{N} [A]_{ji} V_i$, where the $A$ is the adjacency matrix as $[A]_{ji} = (j \rightarrow i)$ and its the eigenvector associated with eigenvalue $V$: $AV = V$.

Table 3 showcases the highest values of closeness centrality, betweenness centrality, and eigenvector centrality in terms of time-frequency connectedness. The findings indicate that the main markets

of varying centrality differ across different dimensions. The carbon market and the energy, metal markets, are considered to be among the markets with the highest centrality. An important observation is that the key markets are primarily concentrated in the energy and metal sectors. Specifically, significant markets in the energy sector include Coal and Oil, while the metal market is dominated by Copper, Gold, and Lead. This finding aligns with the conclusion drawn by Zhou et al.[21], who discovered that the Coal market serves as the central market within the carbon-energy-nonferrous system. At lower frequencies, the carbon trading market exhibits a relatively weaker position or influence. While, the carbon market has become more important in carbon-energy-metal systems during the COVID-19 epidemic period.

## Robustness check

We investigate the reliability of the estimated spillover measures by changing the selection of rolling-window sizes. In order to achieve this goal, Supplementary Fig. 6 depicts the dynamic total volatility spillover index, employing three different rolling window lengths (150, 200, and 250 days), with 200 days being utilized as the baseline in our empirical analysis. The visual examination of these graphs elucidates that the estimation of the dynamic total volatility spillover index remains qualitatively and quantitatively unchanged, irrespective of the chosen rolling window size. These findings lend validation to our results. In order to ensure brevity, the outcomes pertaining to the spillovers of skewness and kurtosis across various rolling window sizes are not shown. However, it is worth noting that the empirical findings obtained with alternative window lengths do not differ significantly from the results of our initial analysis.

## Asymmetric effects of climate risk on the spillovers

Climate change is widely recognized as an important source of risk in the financial system. Since the quantile regression model ignores the possibility of different states of the explanatory variables, and can not reveal the complexity of the influence of independent variables on the dependent variable, we employ the quantile-on-quantile approach (QQ) to examine the relationship between climate risk and spillover effects across carbon, energy and metal markets under different market conditions. The QQ, in fact, provides us with a comprehensive view of asymmetry between variables, enabling more accurate investor decisions and policy advice. Referring to Ding et al.[31], our primary focus is to investigate the influence of climate risk on the overall spillover indices.

Figures 5 and 6 display the response coefficients between the Physical Risk Index (PRI), the Transition Risk Index (TRI), and the total spillover index. Both Fig. 5 and Fig. 6 show that the estimated coefficients vary between different quantiles, and the influence of physical risk and transition risk on the spillover index is asymmetric, that is, its influence on both sides of the tails of the spillover index is different. The impact of physical risk on the total spillover upward risk is higher than the downside risk, especially in long-term spillovers. Transition risk has a higher impact on the upward risk of the total volatility and skewness spillovers, but it has a greater impact on the downside risk of the total kurtosis connectedness.

Second, it is worth noting that in the short run, physical risk mainly has a positive impact on the total spillover index, while in the long run, physical risk negatively affects the total spillover index. This finding can be attributable to the heightened focus of investors for environmental protection in response to climate events, resulting in a short-term decrease in carbon emissions and an increase in the demand for metals[31], increasing the connectedness of carbon-energy-metal system, while this effect will gradually weaken as time goes on. However, the influence of transition risk on total volatility and skewness spillover index at the high- and low-frequency band is almost negative at each quantile level which is contrary to the effect on the kurtosis connectedness. This may be because the transition risk is mainly focused on the implementation of climate-related policies which is conducive to

**Table 3 | Network centrality characteristics of time-frequency spillovers**

| | | Full-sample | | | Pre-COVID-19 | | | During-COVID-19 | | |
|---|---|---|---|---|---|---|---|---|---|---|
| | | Closeness centrality | Betweenness centrality | Eigenvector centrality | Closeness centrality | Betweenness centrality | Eigenvector centrality | Closeness centrality | Betweenness centrality | Eigenvector centrality |
| Net-pairwise spillover | Volatility | Oil | Copper | Coal | EUA | Oil | Copper | EUA | Silver | EUA |
| | Skewness | Lead | Coal | Silver | Gold | Copper | Zinc | Gold | EUA | Coal |
| | Kurtosis | Lead | EUA | EUA | Lead | Gold | Gold | EUA | Silver | EUA |
| Short-term | Volatility | Gold | Oil | Copper | Gas | Oil | EUA | Tin | EUA | Lead |
| | Skewness | Coal | Gold | Gas | Lead | Gold | Gold | Gas | EUA | Coal |
| | Kurtosis | Silver | Coal | Oil | EUA | Oil | Gold | Gold | Lead | Nickel |
| Long-term | Volatility | Oil | Gas | Gold | Lead | Gas | Gas | Lead | Silver | Gold |
| | Skewness | Gold | Lead | Silver | Gas | EUA | Tin | Gold | Gas | Nickel |
| | Kurtosis | Lead | Coal | Oil | Lead | Gold | Coal | Oil | Gold | Lead |

Only the markets with the greatest centrality in different cases are shown in the table. "Net-pairwise spillovers" is the aggregate net-pairwise directional spillovers of the DY model. "Short-term" and "Long-term" is the net-pairwise directional spillover in the short and long-term horizons of BK model, respectively.

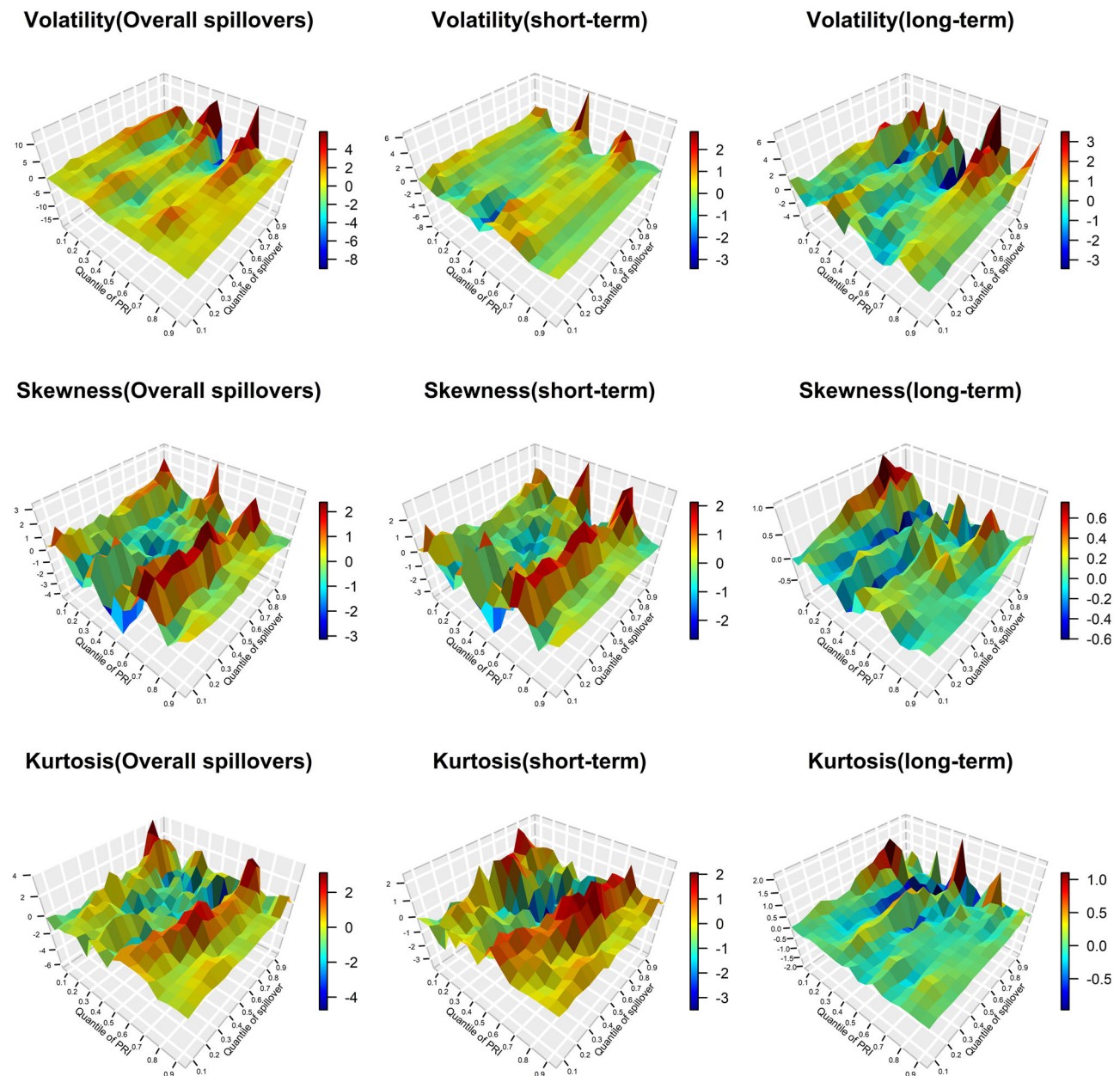

**Fig. 5 | The impact of climate physical risk (PRI) on total spillovers (TCI).** The z-axis represents the estimation of the slope $\beta_1(\theta,\tau)$ (see Eq. 19) which captures the effect of the $\tau$ th quantile of PRI on the $\theta$ th quantile of TCI. The colors in the color bar on the right represent the strengths of correlations. "Overall spillovers" is the dynamic total spillover index of the DY model. The "short-term" and "long-term" is the dynamic frequency connectedness on band:3.14 to 0.14 and 0.14 to 0 of BK model, respectively. Top panels show the volatility, skewness, and kurtosis spillover effects in different frequencies, such as "Volatility (short-term)" represents the coefficients got by the impact of PRI on TCI in the short-term horizon based on QQ model.

the better development of the carbon and the metal market, effectively hedging carbon-energy-metal volatility and skewness spillover effect. Additionally the transition risk may lead to the extreme risk for the market, making a positive impact on the kurtosis connectedness.

Finally, the hedging effects of physical risk and transition risk on volatility, skewness, and kurtosis spillovers are different. The PRI exhibits negative effects on the volatility spillover index, but a more positive effect on the skewness and kurtosis spillover indices between 0.3 and 0.4 quantiles. In terms of volatility risk spillover effects, the TRI has a negative impact on the downside risk, and mainly has a positive impact on the extreme upside risk. However, the total skewness and kurtosis spillover indices are mainly negatively and positively affected by the TRI in each state. It is noteworthy to mention that PRI exerts a positive impact on the volatility, skewness and kurtosis spillover

indices when $\tau$ reaches approximately 0.55 to 0.65, indicating that the losses caused by general climate events may increase the connectedness of the carbon-energy-metal system. The positive effect of PRI on the total spillover index is not significant at 0.9 and 0.95 quantiles, which may be because the duration of extreme climate events is short which is not enough to break the chain of infection between the physical risk and carbon-energy-metal system. The above results are of great importance to the regulatory authorities. In addition to monitoring extreme climate events, we should also be alert to the repeated impact of climate events and the introduction of climate-related policies on other related markets, and establish a regulatory system for the climate risk process.

In summary, the impacts of climate-related physical risk and transition risk on the total spillover indices of the carbon-energy-metal

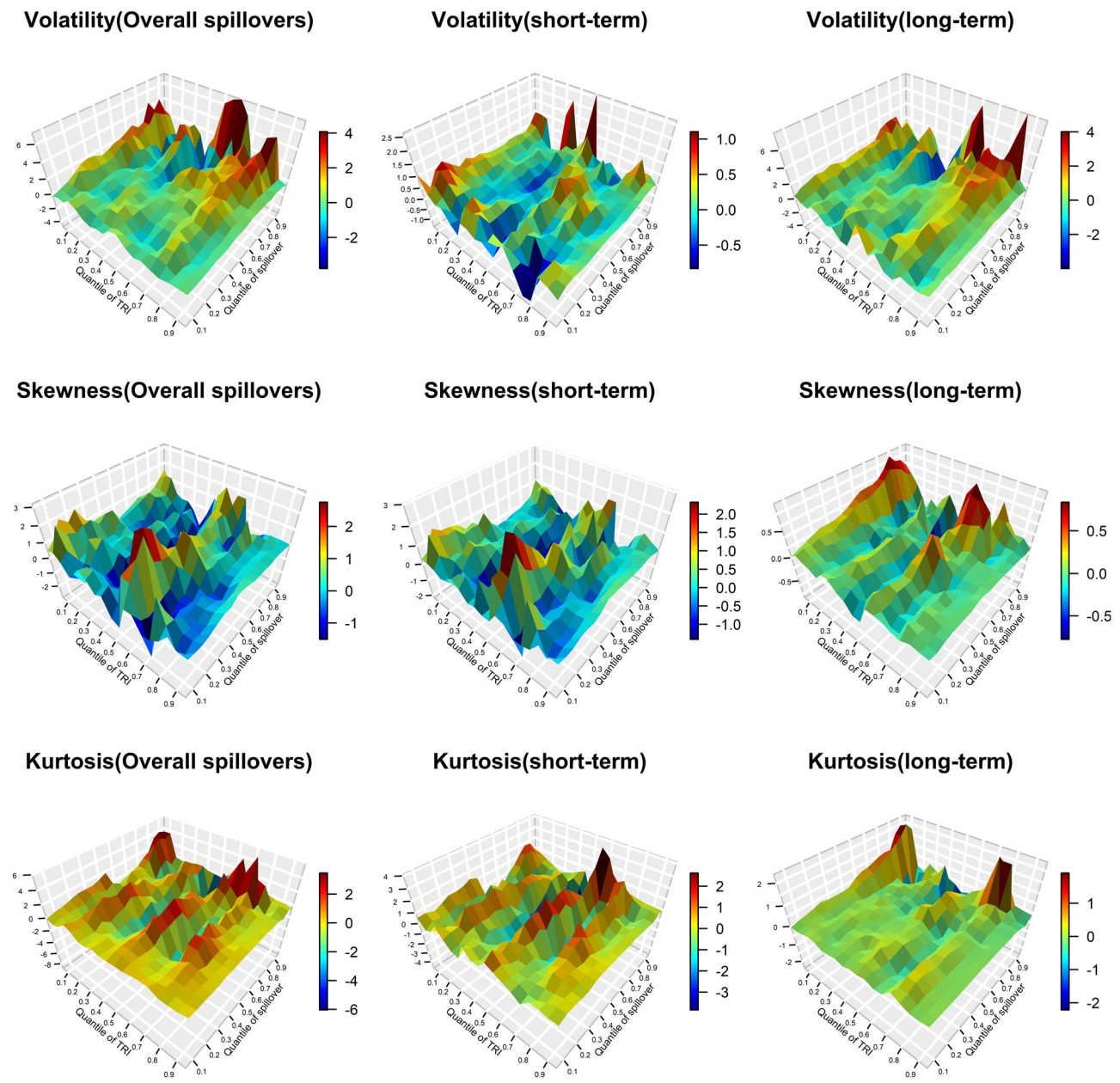

**Fig. 6 | The impact of climate transition risks (TRI) on total spillovers (TCI).** The $z$-axis represents the estimation of the slope $\beta_1(\theta,\tau)$ (see Eq. 19) which captures the effect of the $\tau$ th quantile of TRI on the $\theta$ th quantile of TCI. The colors in the color bar on the right represent the strengths of correlations. "Overall spillovers" is the dynamic total spillover index of the DY model. The "short-term" and "long-term" is the dynamic frequency connectedness on the band:3.14 to 0.14 and 0.14 to 0 of BK model, respectively. The top panels show the volatility, skewness, and kurtosis spillover effects in different frequencies, such as "Volatility (short-term)" represents the coefficients obtained by the impact of TRI on TCI in the short-term horizon based on QQ model.

system are different and have asymmetric characteristics. Physical risk has a higher impact on the total spillover upward risk than the downside risk. This could be because physical losses triggered by extreme climate events bring about greater financial market risks. However, the climate transition risk has a greater impact on the downside risk of the total kurtosis connectedness. This suggests that when there is a high skewness risk, the market's response to climate policy transitions and other related information is insufficient. The coordination between physical risk and total spillover index in the short term is more positive, while the effects of TRI on TCI in the long term are mainly negative. However, the impact of transition risk on volatility and skewness spillovers at high and low frequency appears to be predominantly negative, which contradicts the effect on kurtosis spillovers. It is worth noting that even when both physical risks and

transition risks are at normal levels, they can still have an impact on the cross-market risk spillover effects. This finding echoes the results reported by Mao et al.[50].

To enhance the robustness of our findings, we conduct an additional test by comparing the results obtained through the quantile-on-quantile (QQ) method with those obtained from the quantile regression (QR) method. For comparative analysis, we have selected the estimated Quantile Regression (QR) parameters and compared them with the $\tau$-averaged QQ regression parameters. The equation can be represented as follows:

$$\gamma_1(\theta) = \frac{1}{S}\sum_{\tau}\hat{\beta}_1(\theta,\tau) \tag{3}$$

Where $S = 19$ is the number of quantiles, ranging from 0.05 to 0.95, with an interval of 0.05. The results are shown in Supplementary Figs. 7 and 8. The general patterns of coefficients for climate risk based on the QQ and QR methods demonstrate minor differences, except for the extreme quantiles. This observation further confirms the efficacy of the QQ model. Comparing the results obtained from the QQ and QR models, it can be observed that the parameters of the quantile regression exhibit smaller fluctuations compared to those of the QQ model. This finding validates the results reported by Umar et al.[51]. Both the QQ and QR models show a consistent trend in the impact of climate physical risk on volatility spillover. However, there may be instances of inconsistency, as also documented in previous studies[52]. Despite this, both the QQ and QR models indicate that when climate physical risk and transition risk levels are elevated, the influence of these risks on total risk spillover effects across carbon, energy, and metal markets also intensifies. It is crucial to acknowledge that even in times of moderate climate risk, there is a substantial impact on the long-term skewness risk connectivity.

### Policy implication

Insights gleaned from the findings in this study are useful for both investors and market regulators. These findings hold significant implications for investors who are interested in constructing diversified investment portfolios, as well as for regulators who are seeking to establish climate risk regulatory policies. On the one hand, this study offers a different perspective on the analysis of the carbon-energy-metal system, thereby enriching the existing research on risk spillover effects. On the other hand, the findings have implications for both governments and investors, the specific policy implications can be outlined as follows.

First, from the theoretical perspective, the previous relevant literature mainly focused on the return and volatility level, ignoring the influence of tail series. Therefore, measuring the connectedness of carbon-energy-metal systems should be focused not only on the return and volatility risk spillover effects but also on the high-order moments, further obtain more multidimensional comprehensive research conclusions.

Second, the regulatory authorities should improve their risk management efficiency and guard against the risk contagion across carbon, energy, and metal markets, particularly in times of the outbreak of major emergencies (E.g., SARS, GFC, COVID-19, natural disasters et al.). They should also carefully consider long-term volatility, short-term skewness and kurtosis risk spillover effects. Investors and institutions should pay more attention to extreme risk events and their compound risks (e.g., the compound event and climate risk[53], COVID-19 and climate risk[54,55]). When making investment plans, it is suggested to set up a reasonable investment portfolio to avoid or hedge the adverse effects that may be caused by relevant extreme events in different periods.

Third, the government may consider the formulation of emergency plans in the case of a climate event in order to mitigate the impact of climate risk on the carbon-energy-metal system. Considering that climate change is a global threat, it has a great impact on countries around the world[56]. Hence, it is imperative for governments to enhance communication and coordination of policies across countries, for example, promoting the development of a joint climate convention to reduce extreme climate events (E.g., United Nations Framework Convention on Climate Change, 1992; The Paris Agreement, 2021). In the future, the portfolio construction and optimization of hedging strategies under the framework of high-moment risk spillover can be further discussed.

## Methods
### Data

To determine the volatility, skewness, and kurtosis time-frequency risk spillover effects among carbon, energy and metal markets, We take the prices of WTI oil futures contracts (Oil), natural gas futures (Gas) and Rotterdam coal futures (Coal) launched by London International Petroleum Exchange (IPE), COMEX gold (Gold), COMEX silver (Silver), LME copper futures (Copper), LME aluminum futures (Aluminum), LME Lead futures (Lead), LME Zinc futures (Zinc), LME Nickel futures (Nickel) and the ICE European emission allowance futures (EUA) from the Wind database[21,57]. Since the continuous price data of IPE natural gas futures in 2014 are lacking in records[58], our sample spans July 1, 2015 to June 30, 2022. We select the time period in which both data are recorded. Due to the outbreak of the COVID-19 epidemic and the Russia-Ukraine conflict, the Oil and Nickel market has singular values on April 20 and 21,2020, March 7,2022, respectively. In order to avoid the analysis of the results, the singular values of the three days were specially deleted. Moreover, the returns of the 12 assets are computed as $100 \times (P_t - P_{t-1})/P_{t-1}$ with a total of 1653 observations for each series. In order to study the impact of climate risk, we use the climate risk indices of Bua et al.[59], revised on daily trading hours, to reflect the European climate risks. Compared to other related indicators, these indicators can explore the different effects of physical risk (PRI) and transition risk (TRI) on the total spillover effects, respectively.

Supplementary Table 5 shows the sample minimums (Min.), medians, means, maximums (Max.), skewness, kurtosis, standard deviations (Std. Dev), Jarque-Bera (J-B) tests for normality, Augmented Dickey-Fuller (ADF) tests for stationarity and ARCH-LM test for the ARCH effect of the twelve returns series of carbon, energy and metals markets. According to the mean or the standard deviation of each return series, the returns dispersion is the highest for the Gas, followed by Oil and EUA, and the return of carbon and energy markets appears bigger than that of metal markets which is consistent with the results of Zhou et al.[21]. As observed for skewness coefficient, all the return distribution is asymmetric. All the kurtosis coefficients are not equal to three and the most of return series show peak distribution. The J-B and ADF statistics of all the series reject the null hypotheses, which indicate that the twelve returns series are all stationary and not normally distributed at the 1% significance level. According to the results of ARCH-LM test, the return sequence of each market has a significant ARCH effect which shows that establishing GJRSK model is reasonable.

The heat map which reflects the pairwise correlation between pairs of assets is shown in the Supplementary Fig. 9. We find that the correlation coefficient between the metal markets is relatively high, especially between the Gold and Silver markets. Whereas, the correlation coefficient between carbon and metals markets is lower. This is justifiable given that Gold and Silver belong to a similar.

### Higher-order moment risk measure

The financial time series is not strictly subject to normal distribution, but has the characteristics of leptokurtosis, fat-tail and leverage effect. To measure the conditional volatility, skewness, and kurtosis of carbon, energy and metal markets, we use the GJRSK model proposed by Nakagawa and Uchiyama[36] which is the GARCHSK model with a leverage effect to establish a high-order moment model. The GJRSK model is based on the GJR framework, which allows for asymmetric responses to positive and negative shocks. This is useful for carbon, energy and metals time series which have asymmetric properties[60,61]. The GJRSK model can be expressed as:

$$\begin{cases} r_t = a_1 r_{t-1} + \varepsilon_t \\ h_t = \beta_0 + \beta_1 \varepsilon_{t-1}^2 + \beta_2 h_{t-1} + \beta_3 \varepsilon_{t-1}^2 I_{\{\eta_{t-1}<0\}} \\ s_t = \gamma_0 + \gamma_1 \eta_{t-1}^3 + \gamma_2 s_{t-1} + \gamma_3 \eta_{t-1}^3 I_{\{\eta_{t-1}<0\}} \\ k_t = \delta_0 + \delta_1 \eta_{t-1}^4 + \delta_2 k_{t-1} + \delta_3 \eta_{t-1}^4 I_{\{\eta_{t-1}<0\}} \\ \eta_t = h_t^{-1/2} \varepsilon_t \end{cases} \quad (4)$$

Where $\eta_t|I_{t-1} \sim g(0,1,s_t,k_t)$, and $I_{t-1}$ denotes the information set at time $t$-1. $g(0,1,s_t,k_t)$ is a probability density function with mean 0, variance 1, skewness $s_t$, and kurtosis $k_t$. The parameter of the GJRSK model can be estimated by maximizing the log-likelihood function. The $r_t$ denotes a vector of return of carbon, energy and metals markets and computed by $100 \times (P_t - P_{t-1})/P_{t-1}$, where the $P_t$ is the closing market indices which were obtained on a daily basis.

## Time-frequency connectedness

In order to explore the frequency domain risk spillover effects among carbon, energy and metals markets, we use the frequency connectedness theoretical framework proposed by Baruník and Kehlík[37] to calculate the time-frequency risk spillover effects. According to Diebold and Yilmaz[40], Generalized Forecast Error Variance Decomposition (GFEVD) can be expressed as:

$$(\vartheta_H)_{j,k} = \frac{\sigma_{kk}^{-1} \sum_{h=0}^{H} \left[ (\psi_h \Sigma)_{j,k} \right]^2}{\sum_{h=0}^{H} (\Psi_h \Sigma \Psi_h')_{j,j}} \quad (5)$$

Which, $H$ is the forecast horizon, and $\Psi_h$ is the $n \times n$ order matrix technique. $\sigma_{kk} = (\Sigma)_{k,k}$, $(\vartheta_H)_{j,k}$ measure how much the $k$ th variable contributes to the variance decomposition of the $j$-th element. To make the different $(\vartheta_H)_{j,k}$ comparable, they are standardized, namely:

$$\left(\tilde{\vartheta}_H\right)_{j,k} = \frac{(\vartheta_H)_{j,k}}{\sum_{k=1}^{n} (\vartheta_H)_{j,k}}, \sum_{k=1}^{n} \left(\tilde{\vartheta}_H\right)_{j,k} = 1 \quad (6)$$

Following Diebold and Yilmaz[40], the total connectedness index (TCI) measures the contribution of spillovers of volatility shocks to the system's forecast error variance. Then, the connectedness is defined as the share of the predicted variance generated other than the prediction error itself, namely, the ratio of the sum of the off-diagonal elements to the sum of the entire matrix elements:

$$C(H) = 100 \times \frac{\sum_{\substack{j,k=1 \\ j \neq k}}^{N} \left(\tilde{\vartheta}_H\right)_{j,k}}{\sum_{j,k=1}^{N} \left(\tilde{\vartheta}_H\right)_{j,k}} \quad (7)$$

Where $Tr\{\bullet\}$ is the Trace Operator, $C(H)$ measure the total spillover effect strength of the carbon-energy-metal system. Next, this method measures the size of the spillover effect of the market $k$ on all the remaining markets and the magnitude of market $k$ accepting the spillover effects of all remaining markets, namely, "the total spillover effect on other markets (TO)" and "the total spillover effects from other markets (FROM)", which can be calculated by Eqs. (8) and (9), respectively.

$$C_{\cdot k}(H) = 100 \times \frac{\sum_{\substack{j=1 \\ j \neq k}}^{N} \left(\tilde{\vartheta}_H\right)_{j,k}}{\sum_{j=1}^{N} \left(\tilde{\vartheta}_H\right)_{j,k}} \quad (8)$$

$$C_{k\cdot}(H) = 100 \times \frac{\sum_{\substack{j=1 \\ j \neq k}}^{N} \left(\tilde{\vartheta}_H\right)_{k,j}}{\sum_{j=1}^{N} \left(\tilde{\vartheta}_H\right)_{k,j}} \quad (9)$$

Finally, we obtain the net spillover index which can be obtained by the difference between TO and FROM (Eq. 10), and net pairwise spillover index which can be computed by the difference between total volatility shocks transmitted from market $j$ to $k$ and total volatility shocks transmitted from $k$ to $j$ (Eq. 11).

$$C_k(H) = C_{\cdot k}(H) - C_{k\cdot}(H) \quad (10)$$

$$C_{j,k}(H) = 100 \times \left( \frac{\left(\tilde{\vartheta}_H\right)_{j,k}}{\sum_{k=1}^{N} \left(\tilde{\vartheta}_H\right)_{j,k}} - \frac{\left(\tilde{\vartheta}_H\right)_{k,j}}{\sum_{j=1}^{N} \left(\tilde{\vartheta}_H\right)_{k,j}} \right) \quad (11)$$

Furthermore, we adopt the BK model to study the spectral representation of variance decomposition. Considering the spectral behavior of each sequence, it can be represented by the following Frequency Response Function[62]:

$$S_X(w) = \sum_{h=-\infty}^{\infty} E(X_t X_{t-h}) e^{-iwh} = \varphi(e^{-iw}) \Sigma \varphi'(e^{+iw}) \quad (12)$$

Where $i = \sqrt{-1}$, and $\varphi(e^{-iw}) = \sum_{h=0}^{\infty} \varphi_h e^{-iwh}, h = 1,2,\cdots,H$. The $w$ represents frequency. Power Spectrum $S_X(w)$ which describe how the sequences are distributed over the frequency component $w$ is important to characterize the frequency dynamics. Given a specific frequency $w = \in (-\pi, \pi)$, generalized causation spectrum can be defined as:

$$[f(w)]_{j,k} = \frac{\sigma_{kk}^{-1} \left| (\varphi(e^{-iw}) \Sigma)_{j,k} \right|^2}{[\varphi(e^{-iw}) \Sigma \varphi'(e^{+iw})]_{j,j}} \quad (13)$$

Where, $\varphi(e^{-iw})$ represents the Fourier Transform of pulse effect function $\Psi$, and $[f(w)]_{j,k}$ represents the spectral part of the first variable on the frequency $w$ caused by the shock of the $k$-th variable. According to Baruník and Krehlík[37], given an arbitrary frequency band: $d = (a,b)$ and $a,b \in (-\pi,\pi)$, the total connectedness index under the frequency band $d$ can be specified as:

$$C_d^w = 100 \times \left( 1 - \frac{Tr\left(\{\tilde{\vartheta}_d\}\right)}{\sum \left(\tilde{\vartheta}_d\right)_{j,k}} \right) \quad (14)$$

And, the Frequency Connectedness on the frequency band $d$ can be defined as:

$$C_d^F = 100 \times \left( \frac{\sum_{j \neq k} \left(\tilde{\vartheta}_d\right)_{j,k}}{\sum \left(\tilde{\vartheta}_\infty\right)_{j,k}} - \frac{Tr\left(\{\tilde{\vartheta}_d\}\right)}{\sum \left(\tilde{\vartheta}_\infty\right)_{j,k}} \right) \quad (15)$$

## Quantile-on-quantile regression

Since the quantile regression model ignores the possibility of different states of the explanatory variables, and it cannot reveal the complexity of the influence of independent variables on the dependent variable. Sim and Zhou[63] proposed quantile-on-quantile approach, which is a generalization of the standard quantile regression model, and it combines quantile regression and non-parametric techniques to explore how the quantiles of independent variables affect the conditional quantiles of the dependent variable. Letting the $\theta$ superscript denote the quantile of the TCI, we first postulate a model for the $\theta$-quantile of the TCI ($TCI_t$) as a function of climate risk ($Climate_t$) as:

$$TCI_t = \beta^\theta (Climate_t) + \alpha^\theta TCI_{t-1} + v_t^\theta \quad (16)$$

Where $v_t^\theta$ is an error term that has a zero $\theta$-quantile. The above model can study the spillover effect of climate risk on different quantiles of total connectedness index, but it cannot explain the differentiated effects of different states of climate risk on total spillover effects. High and low climate risk states may have different effects on

connectedness, and connectedness may respond differently to climate risk. Therefore, it is necessary to examine the relationship between the $\tau$ quantile of climate risk and $\theta$ quantile of spillover effects. As $\beta^{\theta}(\bullet)$ unknown, $\beta^{\theta}(Climate_t)$ can be approximated by a first-order Taylor expansion, as follows:

$$\beta^{\theta}(Climate_t) \approx \beta^{\theta}(Climate^{\tau}) + \beta^{\theta'}(Climate^{\tau})(Climate_t - Climate^{\tau}) \quad (17)$$

Rewrite $\beta^{\theta}(Climate^{\tau})$ and $\beta^{\theta'}(Climate^{\tau})$ to $\beta_0(\theta,\tau)$, $\beta_1(\theta,\tau)$, and the above formula is transformed to:

$$\beta^{\theta}(Climate_t) \approx \beta_0(\theta,\tau) + \beta_1(\theta,\tau)(Climate_t - Climate^{\tau}) \quad (18)$$

Replacing formula (18) to the formula (16), get the following formula:

$$TCI_t = \beta_0(\theta,\tau) + \beta_1(\theta,\tau)(Climate_t - Climate^{\tau}) + \alpha(\theta)TCI_{t-1} + \upsilon_t^{\theta} \quad (19)$$

Where $\beta_0(\theta, \tau)$ and $\beta_1(\theta, \tau)$ are the coefficients to be estimated. Unlike the standard quantile regression model, $\beta_0$ and $\beta_1$ are related to $\theta$ and $\tau$ which can capture the impact of $\tau$ quantiles of climate risk on $\theta$ quantiles of the total connectedness index.

## Data availability

The original data used in this study are available at the figshare database (https://figshare.com/articles/dataset/original_data/24033399). All the data except the climate risk indices are made available to the public. The climate risk data used in this paper, namely the Physical Risk Index (PRI) and Transition Risk Index (TRI) of Bua et al.[59] will be, in fact, made available to the public only with the publication of the paper that originates them, namely "Transition Versus Physical Climate Risk Pricing in European Financial Markets: A Text-Based Approach" (corresponding author Lavinia Rognone, email lrognone@ed.ac.uk).

## Code availability

The code used in this study are available at the figshare database (https://figshare.com/articles/dataset/All_code_R/24033336). All other processed data and results are presented in the paper and can be generated through the code.

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

## Acknowledgements

This work is based upon work supported by the Chongqing Municipal Social Science Planning Project under Grant No. 2023NDQN22 (to Y.Z.), the Humanities and Social Science Project of Chongqing Municipal Education Commission under Grant No.23SKGH097 (to Y.Z.), the Fund Project of Chongqing Normal University under Grant No. 19XWB017 (to Y.Z.), the National Social Science Foundation of China under Grant No. 22FGLB075 (to S.W.), the University Philosophy and Social Science Research Project of Jiangsu Province under Grant No. 2021SJA0284 (to S.W.), the University Nature Science Research Project of Jiangsu Province under Grant No. 22KJB630007 (to S.W.), the Development Research Center for Sichuan Petroleum and Natural Gas under Grant No. SKB21-03 (to S.W.), the National Natural Science Foundation of China under Grant No.72204250 (to Z.L.), the Humanities and Social Science Foundation of the Ministry of Education in China under Grant No.21YJCZH093 (to Z.L.), the Social Science Foundation of Jiangsu Province under Grant No. 22GLC022 (to Z.L.), and the Fundamental Research Funds for the Central Universities under Grant No. 2023SK05 (to Z.L.).

## Author contributions

Y.Z. performed the spillover effects calculations, analyzed the data, and prepared the manuscript. S.W. plotted the network, solved the QQ model, and analyzed the data. Z.L. performed the network analysis. L.R. calculated the climate physical risk index and the transition risk index. All authors contributed to writing and reviewing the manuscript.

## Competing interests
The authors declare no competing interests.
