## [Peer Review File · Nature Communications]

The asymmetric effects of climate risk on higher-moment connectedness among carbon, energy and metals marketsREVIEWER COMMENTS

Reviewer #1 (Remarks to the Author):

A. Overview

I thank the editor for inviting me to review the paper entitled "Climate risk and higher moments time-frequency connectedness among carbon, energy and metals markets" which is really on a very good topic. This is an interesting paper examining the effect of climate physical risk and transition risk on the variance, skewness and kurtosis time-frequency risk spillover effects among carbon, energy and metals markets. The paper has been written nicely and I enjoyed reading it. I am recommending for minor revision. Here are my comments.

B. Comments

1. Please clarify why you choose this network layout instead of others in the Fig.7.
2. Please also update references where needed, appropriate, or to enhance relevance with our readership. The following might be relevant depending on your closer scrutiny :
 - Bouri, E., Lei, X., Xu, Y., Zhang, H., 2023. Connectedness in implied higher-order moments of precious metals and energy markets. *Energy*, 263, 125588.
 - Chen, Z., Zhang, L., Weng, C., 2023. Does climate policy uncertainty affect Chinese stock market volatility?. *International Review of Economics & Finance*, 84,369-381.
3. In the conclusion, the interesting part of the results should be further discussed with the literature, and policy implication's part is also weak which needs to be strengthened.
4. In order to get more rigorous conclusion, the rolling-windows spillover effects should examine the sensitivity of the estimated spillover measures to the selection of rolling-window sizes.
5. The contribution and innovation of this article can be more prominent in the introduction part.

Reviewer #2 (Remarks to the Author):

Please see my attachment.

Review report for *Climate risk and higher moments time-frequency
connectedness among carbon, energy and metals markets*

Overview

While this study presents DY and BK model to study the connectedness among carbon, energy and metals markets and further discuss effect from climate physical risk and transfer risk, it is interesting and well organized. However, the article needs revisions before being appropriate for publication. Please see my comments below.

Major comments

1. The authors have stated the approaches used as their contributions (DY, BK, QQ method etc.). However, the methods are existing models, and thus, cannot be considered as contributions from this paper.
2. After the first paragraph of this section, the authors should also explain why it is suitable to use a time-frequency connectedness model when examining the spillovers among carbon, energy and metals markets.
3. What is the advantage of GJRSK over other GARCH models? And the estimation results of GJRSK model are not reported.
4. I think one can show more policy implication analyses in the context of COVID-19.
5. I think one can show further heterogeneity analysis for Network analysis with the sub-sample of post-COVID19 and pre-COVID19.
6. Does it indicate that the connectedness system is not very relevant since the results of static spillovers show that there are small effects from or to other variables?
7. I am wondering if the results of QQ are reasonable since the poor fitting of QQR and QR.
8. The data is interesting, but please explain why you use the daily data since the transmission of information among different markets is relatively slower.

Minor comments

1. I believe the authors can shorten the method partly to show the core equations since the authors do not propose most contents.
2. I think more work can be shown related to carbon, energy and metal markets.
3. Please insert the original Figures.
4. There are many formatting errors in the text that need to be corrected, especially the way formulas are written, such as line 195, 224, 236, 282 etc.

Reviewer #3 (Remarks to the Author):

This paper examines connectedness amongst the Carbon-Energy-Metals network and sheds light on the role that Climate Risk plays in this. In particular there is relatively little evidence on the higher moment connectedness thus far. The paper therefore does therefore fill a gap in the literature and provides some new findings. The implications of these findings should also be further elaborated.

Comments:

- 1) The data sample starts from 2015, which is surprising and there is little justification for this. Reliable carbon return data is available since March 2008 and for other variables should be prior to this. For example for climate data Gupta and Pierzdoch (2022) go back daily to c. 2000. Unless there is a compelling reason otherwise the analysis should start from March 2008.
- 2) From Fig. 2, we can see spikes in the variance data and especially in the skewness and kurtosis data. Usually, a transformation of the data is implemented to reduce this. For variance usually the natural log is taken of the standard deviation. I would recommend the authors to examine the impact of transforming their data to smoothing out these spikes on their results.
- 3) Fig. 3 - Fig. 6, we see that there are are spikes in connectedness that seem to occur when there are spikes in the variables. The message then would be that spikes transmit heavily from market to market but during other periods there is little effect. However, the effect in other periods is plausibly reduced, or hidden, because of the presence of such large spikes in the data.
- 4) There should be further discussion and interpretation of Fig. 10.
- 5) In the conclusion you recommend governments to make "more targeted risk prevention measures" it isn't clear what sort of policy can do this and whether this is actually possible. There should be a recommendation for investors / institutions active in these markets to plan for extreme movements in these market prices.

Minor comments

- 1) p9 lines 271-277 - Climate is mis-spelt

Dear Reviewers,

Thank you for your comments about our manuscript entitled “Climate risk and higher moments time-frequency connectedness among carbon, energy and metals markets” (ID: NCOMMS-22-53953). Those comments are all valuable and very helpful for revising and improving our paper, as well as the important guiding significance to our researches. We have studied comments carefully and have made the correction which we hope meet with approval. The main corrections in the paper are marked with blue words, and the responses to the reviewers are as following:

Reviewers' comments:

Reviewer #1 (Remarks to the Author): I thank the editor for inviting me to review the paper entitled "Climate risk and higher moments time-frequency connectedness among carbon, energy and metals markets" which is really on a very good topic. This is an interesting paper examining the effect of climate physical risk and transition risk on the variance, skewness and kurtosis time-frequency risk spillover effects among carbon, energy and metals markets. The paper has been written nicely and I enjoyed reading it. I am recommending for minor revision. Here are my comments.

1. Please clarify why you choose this network layout instead of others in the Fig.7.

Response: Thank you for your constructive and helpful suggestions, which are very helpful to improve the quality of our paper. Based on your suggested elements, we further explain this in the network analysis of time-frequency spillover section, and the details are as follows:

Network analysis of time-frequency spillover part

“The location of the nodes is decided by the algorithm of Fruchterman and Reingold⁸⁰, which is one of the most used layout algorithms⁸¹ and applies an iterative process to select the location of the nodes to minimize the energy of the system. Then, the nodes sharing more connections are located closer to each other.”

References

Fruchterman, T. M. & Reingold, E.M. Graph drawing by force - directed placement. Softw. Pract. Exper. 21, 1129-1164 (1991).

Bales, S. Sovereign and bank dependence in the eurozone: A multi-scale approach using wavelet-network analysis. Int. Rev. Financ. Anal. 83, 102297(2022).

2. Please also update references where needed, appropriate, or to enhance relevance with our readership. The following might be relevant depending on your closer scrutiny:

Bouri, E., Lei, X., Xu, Y., Zhang, H., 2023. Connectedness in implied higher-order

moments of precious metals and energy markets. *Energy*, 263, 125588.

Chen, Z., Zhang, L., Weng, C., 2023. Does climate policy uncertainty affect Chinese stock market volatility?. *International Review of Economics & Finance*, 84,369-381.

Response: Thank you very much for your valuable suggestions. We have carefully reviewed the latest literature and made revisions and improvements to the literature review section. The details are as follows:

Literature review part

“Cui and Maghyereh⁵² examine the interconnectedness of higher-order moment risks between oil and commodity futures, while Zhou et al.²¹ investigate the spillover effects of higher-order moment tail risks among carbon, energy, and non-ferrous metals markets. In contrast to their studies, our research specifically focuses on the time-frequency spillover effects of higher-order moments. Although Bouri et al.⁵³ investigated the connectedness of volatility, skewness, and kurtosis between precious metals and energy markets in both the time and frequency domains, our study delves into the time-frequency spillover effects of higher-order moments across carbon, energy, and metals markets. Additionally, we consider the impact of the leverage effect when calculating volatility, skewness, and kurtosis.

Furthermore, researchers have categorically classified climate risks into physical risk and transition risk. A study conducted by Karmakar et al.⁶³ has demonstrated the significant predictive capability of physical risks for gold volume, particularly at 5- and 22-day-ahead horizons. Moreover, Salisu et al.⁶⁴ have uncovered noteworthy associations between the return volatility of gold and climate risks. Specifically, they found a positive and significant relationship between return volatility and transition risk, while a negative and significant relationship was observed with physical risk.”

References

Cui, J. & Maghyereh, A. Higher-order moment risk connectedness and optimal investment strategies between international oil and commodity futures markets: Insights from the COVID-19 pandemic and Russia-Ukraine conflict. *Int. Rev. Financ. Anal.* 86, 102520 (2023).

Bouri, E., Lei, X., Xu, Y. & Zhang, H. Connectedness in implied higher-order moments of precious metals and energy markets. *Energy* 263, 125588 (2023).

Zhou, Y., Wu, S. & Zhang, Z. Multidimensional risk spillovers among carbon, energy and nonferrous metals markets: Evidence from the quantile VAR network. *Energ. Econ.* 114, 106319(2022).

Karmakar, S., Gupta, R., Cepni, O. & Rognone, L. Climate risks and predictability of the trading volume of gold: evidence from an INGARCH model. *Resour. Policy* 82, 103438 (2023).

Salisu, A. A., Olaniran, A. & Lasisi, L. Climate risk and gold. *Resour. Policy* 82, 103494 (2023).

3. In the conclusion, the interesting part of the results should be further discussed with

the literature, and policy implication's part is also weak which needs to be strengthened.

Response: Thank you very much for your valuable suggestions. We have carefully reviewed the conclusion part. The details are as follows:

Conclusion part

“However, we find that the spillovers of skewness and kurtosis were significantly lower compared to the spillover of volatility. It is worth noting that the spillover of kurtosis is not necessarily lower than the spillover of skewness.

These results diverge from the findings of Bouri et al.⁵³, who noted a weakening of system-wide connectedness as the moment order increases. Furthermore, they found that the level of spillovers in all implied moments is notably higher at lower frequencies. This difference may arise from the distinctive features of conditional and implied higher-order moments, where the latter captures market participants' expectations of future market behavior.⁸⁶

This finding aligns with the conclusion drawn by Zhou et al.²¹, who discovered that the Coal market serves as the central market within the carbon-energy-nonferrous system. However, the roles of various markets in different higher-order moment risk spillovers are inconsistent. For instance, the Oil market serves as a transmitter of volatility risk spillover and also exhibits a susceptibility to skewness and kurtosis risks.

The research conclusion is of great significance to both investors and market regulators. The research conclusion is of great significance to both investors and market regulators. Which has an important reference value for investors to build a diversified investment portfolio and regulators to formulate climate risk regulatory policies. On one hand, it provides a new perspective on the research of carbon-energy-metal system, and enriches the research content of risk spillover effect. On the other hand, the findings have implications for both governments and investors, the specific policy implications are as follows.

First, from the theoretical perspective, the previous relevant literature mainly focused on the return and volatility level, ignoring the influence of tail series. Therefore, measuring the connectedness of carbon-energy-metal systems should be focused not only on the return and volatility risk spillover effects, but also on the high-order moment, further obtain more multidimensional comprehensive research conclusions.

Second, the regulatory authorities should improve their risk management efficiency and guard against the risk contagion among carbon, energy, and metal markets, especially during the outbreak of major emergencies (E.g., SARS, GFC, Covid-19, War, natural disasters et al.) , also should pay more attention to long-term volatility, short-term skewness and kurtosis risk spillover effects. For the investors and institutions, they should pay more attention to extreme risk events and their compound risks (e.g., the compound event and climate risk⁸⁷, COVID-19 and climate risk⁸⁸⁻⁸⁹). When making investment plans, it is suggested to set up a reasonable investment portfolio to avoid or hedge the adverse effects that may be caused by relevant extreme events in different periods.

Third, the government should formulate emergency plans for climate events occurring to prevent the impact of climate risk on the carbon-energy-metal system. Considering that climate risk is a global threat, it has a great impact on countries around the world⁹⁰. Thus, the

governments should strengthen the communication of relevant policies among countries and jointly cope with climate risk events. For example, promoting the development of joint climate convention to reduce extreme climate events (E.g., United Nations Framework Convention on Climate Change, 1992; The Paris Agreement, 2021). In the future, the portfolio construction and optimization of hedging strategies under the framework of high-moment risk spillover can be further discussed.”

References

- Zhou, Y., Wu, S. & Zhang, Z. *Multidimensional risk spillovers among carbon, energy and nonferrous metals markets: Evidence from the quantile VAR network. Energ. Econ. 114, 106319(2022).*
- Bouri, E., Lei, X., Xu, Y. & Zhang, H. *Connectedness in implied higher-order moments of precious metals and energy markets. Energy 263, 125588 (2023).*
- Poon, S. H. & Granger, C.W. *Forecasting volatility in financial markets: A review. J. Econ. Lit. 41, 478-539 (2003).*
- Zscheischler, J. et al. *Future climate risk from compound events. Nat. Clim. Change 8, 469-477 (2018).*
- Phillips, C. A. et al. *Compound climate risks in the COVID-19 pandemic. Nat. Clim. Change 10, 586-588(2020).*
- Ford, J. D, et al. *Interactions between climate and COVID-19. Lancet Planet. Health 6, 825-833(2022).*
- Ozkan, A., Ozkan, G., Yalaman, A. & Yildiz, Y. *Climate risk, culture and the Covid-19 mortality: A cross-country analysis. World Dev. 141, 105412(2021).*

4. In order to get more rigorous conclusion, the rolling-windows spillover effects should examine the sensitivity of the estimated spillover measures to the selection of rolling-window sizes.

Response: Thank you for your suggestion to examine the sensitivity of the estimated spillover measures to the selection of rolling-window sizes in order to obtain more rigorous conclusions regarding the spillover effects. Following your feedback, we have conducted additional analyses to assess the sensitivity of the estimated spillover measures to different rolling-window sizes. By varying the window sizes and comparing the resulting spillover effects, we have gained a deeper understanding of the robustness and stability of our findings. The details are as follows:

Robustness check part

“In order to get more rigorous conclusion, we examine the sensitivity of the estimated spillover measures to the selection of rolling-window sizes. To achieve that, Fig.11 plots the time-varying total volatility spillover index estimates based on for three alternative rolling window lengths (150,

200 and 250 days), with 200 days the baseline in the empirical analysis. It is clear from the visual observation of these graphs that the estimation of the time-varying total volatility spillover index is not affected qualitatively or quantitatively by the selected rolling window size, thus validating our results which given in our paper. For the sake of brevity, the skewness, and kurtosis spillovers for different rolling window sizes are not displayed. In fact, the empirical results for the alternative window lengths are not significantly different from our initial empirical analysis results.”

Fig. 11. Overall volatility spillover index using different rolling-window sizes.

Notes: see notes in Fig.3.

5. The contribution and innovation of this article can be more prominent in the introduction part.

Response: Thank you for your suggestion to make the contribution and innovation of this article more prominent in the introduction part. Following your feedback, we have revised the introduction section to explicitly highlight the contribution and innovation of our research. We have provided a clearer and more focused explanation of how our study adds to the existing knowledge in the field and the novel insights it brings. The details are as follows:

Introduction part

“This study contributes to the literature in several ways. First, we explore the risk spillover effects in the carbon-energy-metals nexus from the perspective of higher-order moment risk, adding to the previous studies, which mainly investigate the spillover effects of returns and volatility. Our results provide evidence that the risk spillover effects at the higher-order moment level cannot be ignored. Second, the high-order moment risk spillovers are decomposed in different frequency domains, and combined with multidimensional network analysis, the high-order moment risk transfer path in the carbon-energy-metals nexus and its heterogeneity in the short-term and long-term frequency domains are characterized. We find that risk spillovers show strong heterogeneity at different frequencies and higher-order moment levels, with the

long-term volatility spillovers accounting for the largest proportion of cross-market volatility spillovers, while high-order moment risk spillovers mainly occur in the short term. Finally, this study examines the impact of climate risk on the risk spillover effects in the carbon-energy-metals nexus via novel measures of physical and transition climate risk proxies obtained from textual analysis. The empirical findings demonstrate that both physical and transition risks positively affect short-term kurtosis spillovers, but have a negative impact on long-term volatility and skewness spillover effects, thus enriching existing climate risk literature.”

Reviewer #2 (Remarks to the Author): While this study presents DY and BK model to study the connectedness among carbon, energy and metals markets and further discuss effect from climate physical risk and transfer risk, it is interesting and well organized. However, the article needs revisions before being appropriate for publication. Please see my comments below. Major comments and minor comments:

1. The authors have stated the approaches used as their contributions (DY, BK, QQ method etc.). However, the methods are existing models, and thus, cannot be considered as contributions from this paper.

Response: We appreciate your professional comments and suggestions. We agree that the methods used in the paper are existing models and not original contributions. We have rephrased our statement to more accurately reflect the specific contributions of our study. The main contribution of our study is to extend the emerging literature on energy finance in a novel direction by exploring the separate roles of physical and transition climate risks as a determinant of the linkages among carbon, energy, and metal markets. As a second novelty, we try to explore the risk spillover effects in the carbon-energy-metals nexus from the perspective of higher-order moment risk, adding to the previous studies, which mainly investigate the spillover effects of returns and volatility. We have made the following revisions:

Introduction part

“This study contributes to the literature in several ways. First, we explore the risk spillover effects in the carbon-energy-metals nexus from the perspective of higher-order moment risk, adding to the previous studies, which mainly investigate the spillover effects of returns and volatility. Our results provide evidence that the risk spillover effects at the higher-order moment level cannot be ignored. Second, the high-order moment risk spillovers are decomposed in different frequency domains, and combined with multidimensional network analysis, the high-order moment risk transfer path in the carbon-energy-metals nexus and its heterogeneity in the short-term and long-term frequency domains are characterized. We find that risk spillovers show strong heterogeneity at different frequencies and higher-order moment levels, with the long-term volatility spillovers accounting for the largest proportion of cross-market volatility spillovers, while high-order moment risk spillovers mainly occur in the short term. Finally, this study examines the impact of climate risk on the risk spillover effects in the carbon-energy-

metals nexus via novel measures of physical and transition climate risk proxies obtained from textual analysis. The empirical findings demonstrate that both physical and transition risks positively affect short-term kurtosis spillovers, but have a negative impact on long-term volatility and skewness spillover effects, thus enriching existing climate risk literature.”

2. After the first paragraph of this section, the authors should also explain why it is suitable to use a time-frequency connectedness model when examining the spillovers among carbon, energy and metals markets.

Response: Thank you for your constructive and helpful suggestions. We agree that it is important to explain why a time-frequency connectedness model is suitable for examining the spillovers among carbon, energy, and metals markets. Therefore, we have added an explanation in the paper to further elaborate on the suitability of a time-frequency connectedness model for analyzing the spillovers among carbon, energy, and metals markets. The details are as follows:

Introduction part

“Meanwhile, the objectives and preferences of market participants differ in the short and long term, leading to variations in spillover effects among carbon, energy, and metal markets at different frequencies²³. The price behaviors of these markets also exhibit various frequencies, ranging from long-term trends to short-term fluctuations. The cap-and-trade principle of the European Union Emissions Trading System affects the short- and long-term price of carbon market trading demand, causing differences in frequency spillover effects³¹⁻³². Furthermore, energy and metal commodity markets have sticky prices in the short term and exhibit complete flexibility in the long term, resulting in different frequency connections with other markets³³. So, we use a time-frequency connectedness model to identify the driving factors of spillovers among these markets, whether it be long-term trends or short-term fluctuations. This information is valuable to investors and policymakers in devising risk management strategies and making informed investment decisions.”

References

- Jiang, W. & Chen, Y. The time-frequency connectedness among metal, energy and carbon markets pre and during COVID-19 outbreak. Resour. Policy 77,102763(2022).*
- Ding, Q., Huang, J. & Zhang, H. Time-frequency spillovers among carbon, fossil energy and clean energy markets: The effects of attention to climate change. Int. Rev. Financ. Anal. 83, 102222 (2022).*
- Adekoya, O. B., Oliyide, J. A. & Noman, A. The volatility connectedness of the EU carbon market with commodity and financial markets in time- and frequency-domain: The role of the U.S. economic policy uncertainty. Resour. Policy 74, 102252(2021).*
- Ortas, E. & Alvarez, I. The efficacy of the European Union emissions trading scheme: Depicting the co-movement of carbon assets and energy commodities through wavelet decomposition. J. Clean. Prod. 116, 40-49 (2016).*

3. *What is the advantage of GJRSK over other GARCH models ? And the estimation results of GJRSK model are not reported.*

Response: Thank you very much for your valuable question. The advantage of GJRSK over other GARCH models lies in its ability to model asymmetry which is commonly observed in financial data. GJRSK model allows for different dynamics in the positive and negative shocks, leading to more accurate estimation of risk measures. Regarding the estimation results of the GJRSK model, we have provided the estimation results in the revised version of the manuscript, the details are as follows:

Methodology and Data part

“The GJRSK model is based on the GJR framework, which allows for asymmetric responses to positive and negative shocks. This is useful for carbon, energy and metals time series which have asymmetric property⁷⁰⁻⁷¹. ”

Empirical results part

Table 3 The parameter estimation for the GJRSK model.

Parameter	EUA	Oil	Gas	Coal	Gold	Silver	Copper	Aluminum	Zinc	Nickel	Tin	Lead	
Mean equation	α_1	-0.04**	0.10***	0.10***	-0.09***	0.00	-0.02	-0.02	-0.03	-0.06***	-0.04**	-0.04**	0.01
Variance equation	β_0	0.40***	0.36***	1.44***	0.04**	0.00	0.04***	0.01	0.23***	0.02	0.21***	0.04**	0.25***
	β_1	0.10***	0.01	0.02	0.02	0.03	0.07***	0.01	0.02	0.01	0.05**	0.16***	0.08***
	β_2	0.07***	0.13***	0.18***	0.00	0.00	0.00	0.01	0.17***	0.01	0.00	0.00	0.00
	β_3	0.82***	0.86***	0.80***	0.97***	0.96***	0.91***	0.98***	0.73***	0.98***	0.89***	0.84***	0.80***
Skewness equation	γ_0	0.02	-0.07***	0.02	-0.01	-0.08***	0.01	0.00	0.03*	0.01	-0.01	-0.08***	-0.01
	γ_1	-0.01	-0.11***	0.00	0.00	0.00	-0.11***	0.01	0.01	-0.01	0.00	0.00	0.01
	γ_2	0.01	0.11***	0.00	0.00	0.00	0.12***	0.02	0.05**	0.00	0.00	0.07***	-0.04**
	γ_3	0.61***	-0.03*	0.81***	0.72***	-0.46***	0.29***	-0.02	0.68***	-0.59***	0.34***	0.01	0.46***
Kurtosis equation	δ_0	1.10***	3.34***	0.55***	3.94***	0.55***	1.93***	3.42***	1.17***	2.07***	3.45***	3.53***	2.22***
	δ_1	0.00	0.00	0.00	0.00	0.00	0.00	0.00	0.01	0.00	0.00	0.00	0.02
	δ_2	0.00	0.01	0.04**	0.01	0.00	0.01	0.01	0.04**	0.03	0.01	0.03	0.00
	δ_3	0.69***	0.01	0.84***	0.12***	0.84***	0.44***	0.00	0.62***	0.37***	0.00	0.01	0.31***

Notes: ***, **, and * denote 1%, 5%, and 10% level of significance, respectively.

References

- Jiang, Y., Jiang, C., Nie, H. & Mo, B. *The time-varying linkages between global oil market and China's commodity sectors: Evidence from DCC-GJR-GARCH analyses*. *Energy* 166, 577-586 (2019).
- Huang, Z., Liang, F., Wang, T. & Li, C. *Modeling dynamic higher moments of crude oil futures*. *Financ. Res. Lett.* 39, 101570 (2021).

4. I think one can show more policy implication analyses in the context of COVID-19.

Response: Thank you for your suggestion regarding including more policy implication analyses in the context of COVID-19. In response to your suggestion, we have conducted further policy implication analyses that specifically focus on the implications of our findings in the context of COVID-19. We have made the following revisions:

Rolling-windows analysis part

“We contrast the net spillover effects before and after the COVID-19 outbreak on 1 January 2020²³. Figure 7 reports the average net total directional connectedness among carbon, energy, and metal markets in the full-sample period and two phases. The absolute net directional connectedness in most markets during the COVID-19 epidemic period is greater than before the outbreak of COVID-19 epidemic, especially the net skewness and kurtosis spillovers. These results suggest that the outbreak of COVID-19 has significantly increased the volatility, skewness, and kurtosis risk spillover effects of carbon, energy and metal market. The impact of COVID-19 on risk spillover is universal. This may be because the negative sentiment caused by COVID-19 epidemic may trigger pessimistic expectations among investors in the market and amplify the spillover effects between markets through asset adjustments. We can also observe the changes in roles for each market in different periods. Volatility in the gold and copper markets was mainly net spillovers before outbreak of COVID-19, but converted to net recipients during COVID-19 epidemic. However, gold and copper markets have positive short-term net volatility connectedness during the COVID-19 outbreak period which is consistent with the results of Jiang and Chen²³. ”

Fig. 7. Net total directional connectedness.

Notes: This figure shows the net total directional connectedness, representing the mean of each period's dynamic net total directional connectedness.

References

Jiang, W. & Chen, Y. The time-frequency connectedness among metal, energy and carbon markets pre and during COVID-19 outbreak. *Resour. Policy* 77,102763(2022).

5. I think one can show further heterogeneity analysis for Network analysis with the sub-sample of post-COVID19 and pre-COVID19.

Response: Thank you for your suggestion regarding conducting further heterogeneity analysis for network analysis with the sub-sample of post-COVID19 and pre-COVID19. Following your suggestion, we have performed additional analysis to examine the heterogeneity within the network by dividing the dataset into sub-samples of post-COVID19 and pre-COVID19. This analysis provides valuable insights into the potential impact of the COVID-19 pandemic on the network dynamics and allows for a more comprehensive understanding of the results. We have included the results of this heterogeneity analysis in the Network analysis of time-frequency spillover part. The details are as follows:

Network analysis of time-frequency spillover part

“Figs. 9 and 10 depict the net pairwise directional spillover in the pre- and during-

COVID-19 periods. The results show that carbon market are more vulnerable to the shocks from other markets in both the short- and long-term. The spillover effects between metal markets and between energy markets is relatively larger, indicating that the metal and energy markets play a more dominant role. It is worth noting that the spillover effect between the carbon market and the energy, metal market has increased significantly during the COVID-19 epidemic period, indicating that carbon market have become more important in carbon-energy-metal systems. ”

Fig. 9. Net-pairwise directional connectedness before the COVID-19 pandemic period.

Notes: see notes in Fig.8.

Fig. 10. Net-pairwise directional connectedness during the COVID-19 pandemic period.

Notes: see notes in Fig.8.

Table 7 Network centrality characteristics of time-frequency spillovers.

		Full-sample			Pre-COVID-19			During-COVID-19		
		Closeness centrality	Betweenness centrality	Eigenvector centrality	Closeness centrality	Betweenness centrality	Eigenvector centrality	Closeness centrality	Betweenness centrality	Eigenvector centrality
Net-pairwise directional spillover	Volatility	Oil	Copper	Coal	EUA	Oil	EUA	EUA	Silver	EUA
	Skewness	Lead	Coal	Silver	Gold	Copper	Zinc	Gold	EUA	Coal
	Kurtosis	Lead	EUA	EUA	Lead	Gold	Gold	EUA	Silver	EUA
Short-term	Volatility	Gold	Oil	Copper	Gas	Oil	EUA	Tin	EUA	Lead
	Skewness	Coal	Gold	Gas	Lead	Gold	Gold	Gas	EUA	Coal
	Kurtosis	Silver	Coal	Oil	EUA	Oil	Gold	Gold	Lead	Nickel
Long-term	Volatility	Oil	Gas	Gold	Lead	Gas	Gas	Lead	Silver	Gold
	Skewness	Gold	Lead	Silver	Gas	EUA	Tin	Gold	Gas	Nickel
	Kurtosis	Lead	Coal	Oil	Lead	Gold	Coal	Oil	Gold	Lead

Notes: Only the markets with the greatest centrality in different cases are shown in the table.

6. Does it indicate that the connectedness system is not very relevant since the results of static spillovers show that there are small effects from or to other variables?

Response: Thank you very much for your thoughtful question. The small static spillover effects does not necessarily mean that the connectedness system is not relevant. The results of this paper imply that in the full sample, approximately 47.37%, 12.08%, and 20.01% of the forecast error variances are due to volatility, skewness, and kurtosis spillovers among different markets,

respectively. The total spillover index size is close to the results of the existing studies. Such as, Wang and Guo⁷³ showed that the total return, and volatility spillover index between carbon and energy markets is 26.721%, and 26.671%, respectively. They also find that carbon market is a minor producer (only 3.236% to others) of spillover effect because it is not developed. Jiang and Chen²³ observed that the total connectedness among carbon, energy, and metal market is 19.76% and 26.88% in the pre-COVID-19 and postCOVID-19 period, respectively.

In addition, the static spillovers only capture the contemporaneous effects of shocks between variables, whereas the time-varying connectedness approach considers the dynamic relationships between variables over time, which may provide a more complete picture of the spillover effects. The static spillover analysis may not be able to capture the spillover effects that occur over longer time horizons, which may be better captured by a time-varying connectedness approach. From the time-changing perspective, it can be found in Figure 4 that each market has a large spillover effect on other markets in some stages.

References

- Wang, Y. & Guo, Z. *The dynamic spillover between carbon and energy markets: new evidence. Energy* 149, 24-33 (2018).
- Jiang, W. & Chen, Y. *The time-frequency connectedness among metal, energy and carbon markets pre and during COVID-19 outbreak. Resour. Policy* 77,102763(2022).

7. I am wondering if the results of QQ are reasonable since the poor fitting of QQR and QR.

Response: Thank you very much for your feedback. Comparing the results obtained from the QQ and QR models, it can be observed that the parameters of the quantile regression exhibit smaller fluctuations compared to those of the QQ model. This could be attributed to the relatively smaller absolute values of the QR parameters compared to the QQ parameters, causing the variability of QR to appear less pronounced in the graphical representation. This observation aligns with the findings reported by Umar et al.⁸⁴. However, both QQ and QR models suggest that when climate physical risk and transition risk are high, the impact of climate physical risk and transition risk on total risk spillover effects among carbon, energy, and metal markets is significantly increased. Furthermore, it is noteworthy that even during periods of moderate climate risk, there is a substantial impact on the long-term skewness risk connectivity. Although the QQ and QR results exhibit a similar trend in the influence of climate physical risk on volatility spillover, there may still be instances of inconsistency, as also identified in prior literature⁸⁵.

References

- Umar Z., Bossman A., Choi S. & Teplova T. *Does geopolitical risk matter for global asset*

returns? Evidence from quantile-on-quantile regression. *Financ. Res. Lett.* 48, 102991 (2022).

Chen, Z., Zhang, L. & Weng, C. Does climate policy uncertainty affect Chinese stock market volatility?. *Int. Rev. Econ. Financ.* 84, 369-381(2023).

8. The data is interesting, but please explain why you use the daily data since the transmission of information among different markets is relatively slower.

Response: We adopt the daily data which is consistent with the existing literature^{23,25,31} mainly for the following reasons:

Firstly, it's true that the transmission of information among different markets can be relatively slower, particularly for markets that are geographically distant or operate in different time zones. This means that it can take time for new information to be reflected in prices and for the relationships between markets to change. Therefore, compared to the high-frequency data, daily data may be sufficient to capture the relevant dynamics between markets.

Secondly, daily data can be more sensitive to changes in the underlying economic fundamentals and news events. So, it can provide a more detailed picture of time-varying spillover effects among the carbon, energy, and metals markets and allows for a more accurate capture of short-term trends and fluctuations which can be useful for traders, investors, and policymakers who need to make informed decisions in real-time.

Thirdly, using daily data can provide more statistical power and precision in estimating parameters of GJRSK model and Quantile-on-Quantile method. With more data points, it is possible to obtain more accurate estimates of the relationships between markets and variables.

Finally, this study employs a time-frequency spillover approach, which considers the relationships between markets at different frequencies of 1-22 days and more than 22 days, and thus has a good response to the transmission of information over monthly and longer periods.

References

Jiang, W. & Chen, Y. The time-frequency connectedness among metal, energy and carbon markets pre and during COVID-19 outbreak. *Resour. Policy* 77,102763(2022).

Chen, J., Liang, Z., Ding, Q. & Liu, Z. Quantile connectedness between energy, metal, and carbon markets. *Int. Rev. Financ. Anal.* 83, 102282 (2022).

Ding, Q., Huang, J. & Zhang, H. Time-frequency spillovers among carbon, fossil energy and clean energy markets: The effects of attention to climate change. *Int. Rev. Financ. Anal.* 83, 102222 (2022).

9. I believe the authors can shorten the method partly to show the core equations since the authors do not propose most contents.

Response: Thank you very much for your review comments. Based on your suggestion, we have made revisions to the methodology section by streamlining the presentation and focusing

on the core equations, thereby providing a clearer exposition of our method. We believe that these modifications will enhance the readability and comprehensibility of the paper. The details are as follows:

Methodology and Data part

“The financial time series is not strictly subject to normal distribution, but has the characteristics of leptokurtosis, fat-tail and leverage effect. To measure the conditional volatility, skewness, and kurtosis of carbon, energy and metal markets, we use the GJRSK model proposed by Nakagawa and Uchiyama³⁶ which is the GARCHSK model with a leverage effect to establish a high-order moment model. The GJRSK model is based on the GJR framework, which allows for asymmetric responses to positive and negative shocks. This is useful for carbon, energy and metals time series which have asymmetric property⁷⁰⁻⁷¹. The GJRSK model can be expressed as:

$$\begin{cases} r_t = \alpha_1 r_{t-1} + \varepsilon_t \\ h_t = \beta_0 + \beta_1 \varepsilon_{t-1}^2 + \beta_2 h_{t-1} + \beta_3 \varepsilon_{t-1}^2 I_{\{\eta_{t-1} < 0\}} \\ s_t = \gamma_0 + \gamma_1 \eta_{t-1}^3 + \gamma_2 s_{t-1} + \gamma_3 \eta_{t-1}^3 I_{\{\eta_{t-1} < 0\}} \\ k_t = \delta_0 + \delta_1 \eta_{t-1}^4 + \delta_2 k_{t-1} + \delta_3 \eta_{t-1}^4 I_{\{\eta_{t-1} < 0\}} \\ \eta_t = h_t^{-\frac{1}{2}} \varepsilon_t \end{cases} \quad (1)$$

Where $\eta_t | I_{t-1} \sim g(0, 1, s_t, k_t)$, and I_{t-1} denotes the information set at time $t-1$.

$g(0, 1, s_t, k_t)$ is a probability density function with mean 0, variance 1, skewness s_t , and kurtosis k_t . The parameter of the GJRSK model can be estimated by maximizing the log-likelihood function.

The r_t denotes a vector of return of carbon, energy and metals markets and computed by $100 \times (P_t - P_{t-1}) / P_{t-1}$, where the P_t is the closing market indices which were obtained on a daily basis.

Since the quantile regression model ignores the possibility of different states of the explanatory variables, and it cannot reveal the complexity of the influence of independent variables on the dependent variable. Sim and Zhou⁷² proposed quantile-on-quantile approach, which is a generalization of the standard quantile regression model, and it combines quantile regression and non-parametric techniques to explore how the quantiles of independent variables affect the conditional quantiles of the dependent variable. Letting the θ superscript denote the quantile of the

TCI, we first postulate a model for the θ -quantile of the TCI (TCI_t) as a function of climate risk

(Climate_t) as:

$$TCI_t = \beta^\theta (\text{Climate}_t) + \alpha^\theta TCI_{t-1} + v_t^\theta \quad (13)$$

Where v_t^θ is an error term that has a zero θ -quantile. The above model can study the spillover effect of climate risk on different quantiles of total connectedness index, but it cannot explain the differentiated effects of different states of climate risk on total spillover effects. High and low climate risk states may have different effects on connectedness, and connectedness may respond differently to climate risk. Therefore, it is necessary to examine the relationship between the τ quantile of climate risk and θ quantile of spillover effects. As $\beta^\theta(\cdot)$ unknown, $\beta^\theta(\text{Climate}_t)$ can be approximated by a first-order Taylor expansion, as follows:

$$\beta^\theta (\text{Climate}_t) \approx \beta^\theta (\text{Climate}^\tau) + \beta^{\theta'} (\text{Climate}^\tau)(\text{Climate}_t - \text{Climate}^\tau) \quad (14)$$

Rewrite $\beta^\theta (\text{Climate}^\tau)$ and $\beta^{\theta'} (\text{Climate}^\tau)$ to $\beta_0(\theta, \tau)$, $\beta_1(\theta, \tau)$, and the above formula is transformed to:

$$\beta^\theta (\text{Climate}_t) \approx \beta_0(\theta, \tau) + \beta_1(\theta, \tau)(\text{Climate}_t - \text{Climate}^\tau) \quad (15)$$

Replacing formula (15) to the formula (13), get the following formula:

$$TCI_t = \beta_0(\theta, \tau) + \beta_1(\theta, \tau)(\text{Climate}_t - \text{Climate}^\tau) + \alpha(\theta)TCI_{t-1} + v_t^\theta \quad (16)$$

Where $\beta_0(\theta, \tau)$ and $\beta_1(\theta, \tau)$ are the coefficients to be estimated. Unlike the standard quantile regression model, β_0 and β_1 are related to θ and τ which can capture the impact of τ quantiles of climate risk on θ quantiles of total connectedness index.”

References

- Nakagawa, K. & Uchiyama, Y. GO-GJRSK Model with Application to Higher Order Risk-Based Portfolio. *Mathematics* 8, 1990 (2020).
- Jiang, Y., Jiang, C., Nie, H. & Mo, B. The time-varying linkages between global oil market and China's commodity sectors: Evidence from DCC-GJR-GARCH analyses. *Energy* 166, 577-586 (2019).
- Huang, Z., Liang, F., Wang, T. & Li, C. Modeling dynamic higher moments of crude oil futures. *Financ. Res. Lett.* 39, 101570 (2021).
- Sim, N. & Zhou, H. Oil prices, US stock return, and the dependence between their quantiles. *J. Bank. Financ.* 55, 1-8(2015).

10. I think more work can be shown related to carbon, energy and metal markets.

Response: Thank you for your suggestion. Based on your suggestion, we have incorporated some updated literature into this article. The details are as follows:

Literature review part

“Cui and Maghyereh⁵² examine the interconnectedness of higher-order moment risks between oil and commodity futures, while Zhou et al.²¹ investigate the spillover effects of higher-moment tail risks among carbon, energy, and non-ferrous metals markets. In contrast to their studies, our research specifically focuses on the time-frequency spillover effects of higher-order moments. Although Bouri et al.⁵³ investigated the connectedness of volatility, skewness, and kurtosis between precious metals and energy markets in both the time and frequency domains, our study delves into the time-frequency spillover effects of higher-order moments across carbon, energy, and metals markets. Additionally, we consider the impact of the leverage effect when calculating volatility, skewness, and kurtosis.”

References

- Cui, J. & Maghyereh, A. Higher-order moment risk connectedness and optimal investment strategies between international oil and commodity futures markets: Insights from the COVID-19 pandemic and Russia-Ukraine conflict. *Int. Rev. Financ. Anal.* 86, 102520 (2023).
- Bouri, E., Lei, X., Xu, Y. & Zhang, H. Connectedness in implied higher-order moments of precious metals and energy markets. *Energy* 263, 125588 (2023).
- Zhou, Y., Wu, S. & Zhang, Z. Multidimensional risk spillovers among carbon, energy and nonferrous metals markets: Evidence from the quantile VAR network. *Energ. Econ.* 114, 106319(2022).

11. Please insert the original Figures.

Response: Thank you very much for your feedback. We have now reinserted the original figures into the manuscript file to ensure the clarity of the graphics. By doing so, we aim to enhance the visual quality of the figures. We believe that including the original figures will significantly improve the overall presentation of our research.

12. There are many formatting errors in the text that need to be corrected, especially the way formulas are written, such as line195, 224, 236, 282 etc.

Response: Thank you very much for pointing out the formatting errors in the manuscript. We have now thoroughly reviewed and revised the formatting of the paper, ensuring that it adheres to the proper guidelines. Additionally, we have carefully checked for any word usage, grammar, and syntax issues and made the necessary corrections. We sincerely appreciate your attention to detail and for bringing these matters to our attention. Your feedback has been invaluable in improving the overall quality of our manuscript.

Reviewer #3 (Remarks to the Author): This paper examines connectedness amongst the Carbon-Energy-Metals network and sheds light on the role that Climate Risk plays in

this. In particular there is relatively little evidence on the higher moment connectedness thus far. The paper therefore does therefore fill a gap in the literature and provides some new findings. The implications of these findings should also be further elaborated. Comments:

1. The data sample starts from 2015, which is surprising and there is little justification for this. Reliable carbon return data is available since March 2008 and for other variables should be prior to this. For example for climate data Gupta and Pierzdiach (2022) go back daily to c. 2000. Unless there is a compelling reason otherwise the analysis should start from March 2008.

Response: Thank you very much for your constructive feedback. We also hope to explore the high-order moment risk spillover effects among the carbon, energy, and metal markets over a longer sample period, as well as the impact of climate risk on spillovers. However, we were only able to obtain reliable data for the carbon and metal markets dating back to 2008, while for the energy market, the continuous price data of IPE natural gas futures in 2014 are lacked in records^{21,73}. To ensure data continuity, this article's sample period starts from 2015. Additionally, the entire sample is composed of daily data and covers typical unexpected events such as the COVID-19 pandemic and the Russia-Ukraine conflict, enabling us to analyze the impact of extreme events on inter-market risk spillovers.

References

- Wang, Y. & Guo, Z. The dynamic spillover between carbon and energy markets: new evidence. Energy 149, 24-33 (2018).*
- Zhou, Y., Wu, S. & Zhang, Z. Multidimensional risk spillovers among carbon, energy and nonferrous metals markets: Evidence from the quantile VAR network. Energ. Econ. 114, 106319(2022).*

2. From Fig. 2, we can see spikes in the variance data and especially in the skewness and kurtosis data. Usually, a transformation of the data is implemented to reduce this. For variance usually the natural log is taken of the standard deviation. I would recommend the authors to examine the impact of transforming their data to smoothing out these spikes on their results.

Response: Thank you for your feedback. In light of the data's multiple peaks, some studies have utilized logarithmic transformation to achieve data smoothing. However, since skewness can exhibit both right and left tail behaviors, it is not suitable for logarithmic transformation, unlike volatility data. Additionally, the presence of peaks indicates the heavy-tailed nature of financial data, accurately reflecting real-world characteristics. Therefore, to ensure consistent analysis and comparison of volatility, skewness, and kurtosis, we did not transform the data in our study. Nonetheless, in response to your suggestion, we have recalculated the results for volatility

spillover, the revised findings are presented below. Notably, the results demonstrate no significant differences between the transformed data and original results reported in the article.

Static variance spillovers (%).

Model		EUA	Oil	Gas	Coal	Gold	Silver	Copper	Aluminum	Zinc	Nickel	Tin	Lead	FROM
Panel A: DY(2012)	EUA	79.65	2.17	3.07	4.38	0.60	1.53	0.33	0.04	0.52	2.60	2.89	2.22	1.70
	Oil	4.13	61.36	0.96	0.72	8.40	3.28	3.58	5.71	4.76	0.29	5.94	0.88	3.22
	Gas	6.97	0.40	57.44	26.12	0.13	0.49	1.24	0.50	4.12	0.95	0.58	1.06	3.55
	Coal	6.96	0.17	11.41	73.20	0.05	0.39	4.25	1.60	1.41	0.17	0.31	0.09	2.23
	Gold	4.44	11.15	0.47	0.45	50.27	26.81	1.33	1.26	0.47	1.22	1.99	0.14	4.14
	Silver	2.35	5.75	0.29	0.92	31.41	54.37	0.49	0.10	1.22	0.75	0.59	1.74	3.80
	Copper	2.66	7.04	0.89	0.78	4.34	12.61	37.25	4.61	6.70	5.42	16.03	1.66	5.23
	Aluminum	2.07	1.50	4.15	7.24	0.96	1.01	6.95	59.35	4.79	6.43	4.48	1.06	3.39
	Zinc	1.00	0.90	4.37	10.13	0.48	1.06	7.99	5.40	41.62	5.83	7.42	13.79	4.87
	Nickel	1.83	0.18	6.53	10.29	1.30	1.15	3.17	4.19	3.47	58.31	8.31	1.28	3.47
	Tin	1.80	2.51	5.29	8.72	3.63	1.57	4.26	2.91	3.89	4.75	58.73	1.94	3.44
	Lead	1.16	1.73	0.78	2.54	1.78	6.45	3.27	1.95	15.36	3.55	3.72	57.71	3.52
	TO	2.95	2.79	3.18	6.03	4.42	4.70	3.07	2.36	3.89	2.66	4.36	2.15	TCI=42.56
Panel B: BK(2018) Frequency 1 (High frequency): 1days to 22 days	EUA	19.87	0.10	0.69	0.15	0.13	0.12	0.10	0.01	0.03	0.06	0.12	0.23	0.14
	Oil	0.08	9.20	0.07	0.00	0.02	0.02	0.62	0.55	0.48	0.04	0.30	0.26	0.20
	Gas	0.14	0.08	14.82	0.11	0.02	0.03	0.08	0.01	0.20	0.03	0.11	0.11	0.08
	Coal	0.06	0.00	0.05	0.92	0.01	0.03	0.22	0.03	0.02	0.00	0.01	0.01	0.04
	Gold	0.03	0.10	0.01	0.01	0.78	0.10	0.01	0.00	0.03	0.02	0.00	0.04	0.03
	Silver	0.00	0.04	0.01	0.04	0.84	5.22	0.15	0.05	0.25	0.21	0.07	0.25	0.16
	Copper	0.01	0.00	0.02	0.06	0.03	0.04	1.20	0.07	0.20	0.09	0.05	0.17	0.06
	Aluminum	0.10	0.98	0.08	0.20	0.44	0.56	1.38	31.36	1.84	0.81	0.64	0.33	0.61
	Zinc	0.01	0.08	0.06	0.50	0.09	0.13	0.07	0.08	1.23	0.04	0.04	0.23	0.11
	Nickel	0.05	0.03	0.25	0.20	0.04	0.31	0.89	0.28	0.52	15.97	0.68	0.15	0.28
Tin	0.01	0.19	0.03	0.09	0.17	0.13	0.57	0.46	1.25	0.61	13.90	0.79	0.36	

	Lead	0.20	0.38	0.10	0.11	0.46	1.70	1.63	0.79	5.65	0.94	1.23	29.34	1.10
	TO_ABS	0.06	0.17	0.11	0.12	0.19	0.26	0.48	0.19	0.87	0.24	0.27	0.21	TCI=3.17
	EUA	59.78	2.07	2.38	4.23	0.47	1.41	0.23	0.03	0.49	2.54	2.77	1.99	1.55
	Oil	4.04	52.16	0.90	0.72	8.38	3.26	2.96	5.15	4.29	0.24	5.64	0.62	3.02
	Gas	6.83	0.31	42.62	26.01	0.11	0.46	1.16	0.49	3.92	0.92	0.47	0.94	3.47
	Coal	6.91	0.17	11.36	72.29	0.04	0.36	4.03	1.57	1.38	0.17	0.30	0.07	2.20
Panel B:	Gold	4.41	11.05	0.45	0.45	49.49	26.71	1.32	1.26	0.44	1.20	1.98	0.10	4.12
BK(2018)	Silver	2.35	5.71	0.29	0.88	30.57	49.16	0.34	0.05	0.98	0.54	0.52	1.49	3.64
Frequency 1	Copper	2.65	7.04	0.87	0.72	4.31	12.57	36.05	4.54	6.51	5.33	15.99	1.49	5.17
(Low frequency):	Aluminum	1.97	0.52	4.07	7.04	0.51	0.45	5.57	27.99	2.94	5.62	3.85	0.73	2.77
22 days to infinity	Zinc	1.00	0.82	4.31	9.63	0.39	0.93	7.92	5.32	40.38	5.79	7.38	13.56	4.75
	Nickel	1.78	0.15	6.28	10.09	1.26	0.84	2.28	3.90	2.95	42.33	7.63	1.13	3.19
	Tin	1.79	2.32	5.25	8.64	3.46	1.44	3.69	2.45	2.64	4.14	44.84	1.15	3.08
	Lead	0.96	1.36	0.68	2.43	1.32	4.75	1.63	1.16	9.71	2.60	2.49	28.37	2.42
	TO_ABS	2.89	2.63	3.07	5.90	4.24	4.43	2.60	2.16	3.02	2.43	4.09	1.94	TCI=39.39

Notes: This table presents the static spillover connectedness based on the DY method and BK model. We provide the total spillover index (denoted by the term “TCI”), the directional spillover received (denoted by “FROM”), and transmitted (denoted by “TO” or “TO_ABS”) by each market. The jkth value is the directional connectedness from k to j.

3. Fig. 3 - Fig. 6, we see that there are are spikes in connectedness that seem to occur when there are spikes in the variables. The message then would be that spikes transmit heavily from market to market but during other periods there is little effect. However, the effect in other periods is plausibly reduced, or hidden, because of the presence of such large spikes in the data.

Response: Thank you for your constructive and helpful suggestions, which are very helpful to improve the quality of our paper. Based on your suggested elements, we have carefully revised the Fig. 3 - Fig. 6, the details are as follows:

Empirical results part

Fig. 3. Dynamic overall spillovers of carbon, energy and metals markets

Notes: In order to avoid extreme values from masking other period trends, we drew the plots with the monthly mean of dynamic total connectedness. “Overall spillovers” is the dynamic total spillover index of the DY model. “Short-term” and “Long-term” is the dynamic frequency connectedness on band: 3.14 to 0.14 and 0.14 to 0 of BK model, respectively.

Fig. 4. Dynamic net volatility directional spillovers

Notes: The plots are drawn by the monthly mean of dynamic net directional spillovers. The black horizontal line represents $y=0$. "Net directional spillovers" is the aggregate net directional spillovers of the DY model. "Short-term" and "Long-term" is the net directional spillover in the short and long-term horizons of BK model, respectively.

Fig. 5. Dynamic net skewness directional spillovers

Notes: see notes in Fig.4.

Fig. 6. Dynamic net kurtosis directional spillovers

Notes: see notes in Fig.4.”

4. There should be further discussion and interpretation of Fig. 10.

Response: Thank you for your constructive and helpful suggestions, which are very helpful to improve the quality of our paper. Based on your suggested elements, we have improved the part of asymmetric effects of climate risk on the spillovers, and the details are as follows:

Asymmetric effects of climate risk on the spillovers part

“Comparing the results obtained from the QQ and QR models, it can be observed that the parameters of the quantile regression exhibit smaller fluctuations compared to those of the QQ model. This finding is similar to the findings reported by Umar et al.⁸⁴. Both the QQ and QR models show a consistent trend in the impact of climate physical risk on volatility spillover. However, there may be instances of inconsistency, as also documented in previous studies⁸⁵. Nevertheless, both QQ and QR models suggest that when climate physical risk and transition risk are high, the impact of climate physical risk and transition risk on total risk spillover effects among carbon, energy, and metal markets is significantly increased. It is important to note that even during periods of moderate climate risk, there is a substantial impact on the long-term skewness risk connectivity.”

References

- Umar Z., Bossman A., Choi S. & Teplova T. Does geopolitical risk matter for global asset returns? Evidence from quantile-on-quantile regression. *Financ. Res. Lett.* 48, 102991 (2022).
- Chen, Z., Zhang, L. & Weng, C. Does climate policy uncertainty affect Chinese stock market volatility?. *Int. Rev. Econ. Financ.* 84, 369-381(2023).

5. In the conclusion you recommend governments to make "more targeted risk prevention measures" it isn't clear what sort of policy can do this and whether this is actually possible. There should be a recommendation for investors / institutions active in these markets to plan for extreme movements in these market prices.

Response: Thank you for your constructive and helpful suggestions, which are very helpful to improve the quality of our paper. After searching for relevant literatures, we found it is difficult to clarify these “more targeted risk prevention measures”, so we modified them to “improve the risk management efficiency of regulatory agencies”. Moreover, we suggest that investors/institutions active in these markets to plan for extreme movements in these market prices, the detailed revisions are shown in the Conclusion part.

Conclusion part

“Second, the regulatory authorities should improve their risk management efficiency and guard against the risk contagion among carbon, energy, and metal markets, especially during the outbreak of major emergencies (E.g., SARS, GFC, Covid-19, War, natural disasters et al.) ,

also should pay more attention to long-term volatility, short-term skewness and kurtosis risk spillover effects. For the investors and institutions, they should pay more attention to extreme risk events and their compound risks (e.g., the compound event and climate risk⁸⁷, COVID-19 and climate risk⁸⁸⁻⁸⁹). When making investment plans, it is suggested to set up a reasonable investment portfolio to avoid or hedge the adverse effects that may be caused by relevant extreme events in different periods.”

References

- Zscheischler, J. et al. Future climate risk from compound events. *Nat. Clim. Change* 8, 469-477 (2018).
- Phillips, C. A. et al. Compound climate risks in the COVID-19 pandemic. *Nat. Clim. Change* 10, 586-588(2020).
- Ford, J. D, et al. Interactions between climate and COVID-19. *Lancet Planet. Health* 6, 825-833(2022).

6. p9 lines 271-277 - Climate is mis-spelt.

Response: Thank you for your careful review of our paper and your valuable feedback. We appreciate your diligence and assistance in ensuring the accuracy of our work. We apologize for the misspelling of "climate" in the manuscript. We have reviewed the mentioned lines (271-277) and corrected the spelling error accordingly. The details are as follows:

Quantile-on-Quantile regression part

“Therefore, it is necessary to examine the relationship between the τ quantile of climate risk and θ quantile of spillover effects. As $\beta^\theta(\bullet)$ unknown, $\beta^\theta(\text{Climate}_t)$ can be approximated by a first-order Taylor expansion, as follows:

$$\beta^\theta(\text{Climate}_t) \approx \beta^\theta(\text{Climate}^\tau) + \beta^{\theta'}(\text{Climate}^\tau)(\text{Climate}_t - \text{Climate}^\tau) \quad (14)$$

Rewrite $\beta^\theta(\text{Climate}^\tau)$ and $\beta^{\theta'}(\text{Climate}^\tau)$ to $\beta_0(\theta, \tau)$, $\beta_1(\theta, \tau)$, and the above formula is transformed to:

$$\beta^\theta(\text{Climate}_t) \approx \beta_0(\theta, \tau) + \beta_1(\theta, \tau)(\text{Climate}_t - \text{Climate}^\tau) \quad (15)$$

Replacing formula (15) to the formula (13), get the following formula:

$$TCI_t = \beta_0(\theta, \tau) + \beta_1(\theta, \tau)(\text{Climate}_t - \text{Climate}^\tau) + \alpha(\theta)TCI_{t-1} + v_t^\theta \quad (16)$$

”

REVIEWERS' COMMENTS

Reviewer #2 (Remarks to the Author):

The authors have carefully addressed my comments and concerns. My recommendation is Accept.

Reviewer #3 (Remarks to the Author):

The authors have done an excellent job in responding to the queries raised in my previous review.

I now recommend the paper be accepted for publication.

I would also like to make the authors aware of a couple of further points.

i) For natural gas, there is a futures based measure available from S&P GSCI which goes back to 1994 at a daily frequency.

ii) For skewness and kurtosis, these could be potentially normalized for example by the variance as in Amaya et al. (2015).

Amaya, D., Christoffersen, P., Jacobs, K., & Vasquez, A. (2015). Does realized skewness predict the cross-section of equity returns?. *Journal of Financial Economics*, 118(1), 135-167.

Dear Reviewers,

Thank you for your comments about our manuscript entitled “Climate risk and higher moments time-frequency connectedness among carbon, energy and metals markets” (ID: NCOMMS-22-53953A). Those comments are very important for our research. We have studied comments carefully, and the responses to the reviewers are as following:

Responses to the reviewers’ comments:

Reviewer #2 (Remarks to the Author): The authors have carefully addressed my comments and concerns. My recommendation is Accept.

Reviewer #3 (Remarks to the Author): The authors have done an excellent job in responding to the queries raised in my previous review. I now recommend the paper be accepted for publication. I would also like to make the authors aware of a couple of further points.

i) For natural gas, there is a futures based measure available from S&P GSCI which goes back to 1994 at a daily frequency.

Response: Thank you very much for your guidance and helpful opinions, which let us have a broader understanding of natural gas market and provide us with more choices for our future research. Considering that this paper not only considers the natural gas market, but also the oil and coal markets in the energy market, the futures launched by IPE are selected, keeping the data source consistent. Meanwhile, WTI oil futures contracts (Oil), natural gas futures (Gas) and Rotterdam coal futures (Coal) launched by IPE are widely used by many scholars(see, e.g., Wang and Guo, 2018; Zhou et al., 2022).

References

- Wang, Y. & Guo, Z. *The dynamic spillover between carbon and energy markets: new evidence. Energy* 149, 24-33 (2018).
- Zhou, Y., Wu, S. & Zhang, Z. *Multidimensional risk spillovers among carbon, energy and nonferrous metals markets: Evidence from the quantile VAR network. Energ. Econ.* 114, 106319(2022).

ii) For skewness and kurtosis, these could be potentially normalized for example by the variance as in Amaya et al. (2015).

Amaya, D., Christoffersen, P., Jacobs, K., & Vasquez, A. (2015). Does realized skewness predict the cross-section of equity returns?. Journal of Financial Economics, 118(1), 135-167.

Response: Thank you very much for your helpful advice, and we rethink the standardization of

skewness and kurtosis. By reviewing more literature(see, e.g., Mei et al., 2017; Gkillas et al., 2019), we found that the same formula was used to calculate the realized skewness and kurtosis when computed by the intraday return. However, The higher order moments which get by the GARCH class model are basically not standardized again (see, e.g., Wu et al., 2020; Zhu et al., 2021). This may be related to the conventional concepts about skewness and kurtosis risks. Considering the GARCH class model used in this paper to estimate higher moments, we did not further standardize them.

References

- Gkillas, K., Gupta, R. & Pierdzioch, C. Forecasting (downside and upside) realized exchange-rate volatility: is there a role for realized skewness and kurtosis?. *Physica A* 532, 121867(2019).
- Mei, D., Liu, J., Ma, F. & Chen, W. Forecasting stock market volatility: do realized skewness and kurtosis help?. *Physica A* 481, 153-159(2017).
- Wu, X., Xia, M. & Zhang, H. Forecasting var using realized egarch model with skewness and kurtosis. *Financ. Res. Lett.* 32,101090(2020).
- Zhu, P., Tang, Y., Wei, Y., Dai, Y. & Lu, T. Relationships and portfolios between oil and Chinese stock sectors: A study based on wavelet denoising-higher moments perspective. *Energy* 217,119416(2021).